# Storylines of the 2018 Northern Hemisphere heat wave at pre-industrial and higher global warming levels

Kathrin Wehrli[1], Mathias Hauser[1], and Sonia I. Seneviratne[1]

[1]Institute for Atmospheric and Climate Science, Department of Environmental Systems Science, ETH Zurich, Zurich, Switzerland

**Correspondence:** Kathrin Wehrli (kathrin.wehrli@env.ethz.ch)

**Abstract.** Extreme temperatures were experienced over a large part of the Northern Hemisphere during the 2018 boreal summer (hereafter referred to as "NH2018 event"), leading to major impacts to agriculture and society in the affected countries. Previous studies highlighted both the anomalous atmospheric circulation patterns during the event and the background warming due to human greenhouse gas emissions as main drivers for the event. In this study, we present Earth System Model experiments investigating different storylines of the NH2018 event given the same atmospheric circulation and alternative background global warming for: no human imprint, the 2018 conditions, and different mean global warming levels ($1.5°C$, $2°C$, $3°C$, and $4°C$). The results reveal that the human-induced background warming was a strong contributor to the intensity of the NH2018 event, and that resulting extremes under similar atmospheric circulation conditions at higher levels of global warming would reach dangerous levels. Compared to $9\%$ during the NH2018 event, about $13\%$ ($34\%$) of the inhabited or agricultural area in the investigated region would reach daily maximum temperatures over $40°C$ under $2°C$ ($4°C$) of global warming and similar atmospheric circulation conditions.

*Copyright statement.* TEXT

## 1 Introduction

During the 2018 boreal summer, large parts of the Northern Hemisphere experienced extreme temperatures, leading to major impacts to agriculture and society in the affected countries (Vogel et al., 2019). The event (hereafter referred to as "NH2018 event") was associated with near-simultaneous heat waves on three continents, including North America, Western and Northern Europe, as well as Japan and the Korean Peninsula (Kornhuber et al., 2019b; Vogel et al., 2019). Previous studies highlighted both the anomalous atmospheric circulation patterns during the event and the background global warming due to human greenhouse gas emissions as main drivers for the event (Drouard et al., 2019; Kornhuber et al., 2019b; Toreti et al., 2019; Vogel et al., 2019). The NH2018 event was characterized by a hemisphere-wide wavenumber 7 circulation pattern, which was also observed during the European heat waves of 2003, 2006 and 2015 (Kornhuber et al., 2019b). A strong positive mode of the North Atlantic Oscillation contributed significantly to the extreme summer conditions in Europe by amplifying the weather

anomalies induced by the wavenumber 7 pattern (Drouard et al., 2019). An analysis of simulations from the 5th phase of the Coupled Model Intercomparison Project (CMIP5, Taylor et al., 2012) showed that the total area affected by hot extremes during the NH2018 event, despite being unprecedented in the historical record, was actually consistent with the present-day level of global warming (Vogel et al., 2019). Indeed, it had approximately a 16% probability of occurrence under present global warming in the CMIP5 simulations (Vogel et al., 2019). However, no studies were conducted so far to disentangle the contribution of the anomalous circulation patterns vs. background global warming for the climate anomalies during the NH2018 event.

In this study, we present numerical experiments investigating different storylines of the NH2018 event given alternative background global warming but the same atmospheric circulation anomalies. While it cannot provide information on probability, the storyline approach allows us to explore the consequences of a specific event for different levels of future climate warming to improve understanding of the involved driving factors (Hazeleger et al., 2015; Shepherd et al., 2018). The alternative background global warming conditions applied in the experiments include: a) no human imprint (i.e. natural/ pre-industrial climate conditions), b) 2018 conditions (corresponds to approximately $1.1°C$ of global warming in the CMIP5 multi-model mean), c) $1.5°C$, d) $2°C$, e) $3°C$ and, f) $4°C$ of global warming. The atmospheric circulation in the experiments is nudged to the observed wind patterns during the NH2018 event following the approach of Wehrli et al. (2018). Hence, all of the experiments include the same circulation patterns but different background global warming. These experiments are of particular relevance since events associated with the type of circulation patterns from the NH2018 events could lead to high risks of crop failures across several breadbasket regions of the world (Kornhuber et al., 2019a).

## 2 Model and methods

Global climate model simulations are conducted with the Community Earth System Model version 1.2 (CESM, Hurrell et al., 2013). Historical sea surface temperatures (SSTs) and sea ice fractions are prescribed using transient monthly observations from a merged product combining the Hadley Centre sea ice and SST data set, version 1 (HadISST1) up to 1981 with the weekly optimum interpolation SST analysis version 2 by the National Oceanic and Atmospheric Administration (Hurrell et al., 2008, NOAA OIv2 hereafter). We produce SSTs and sea ice consistent with the different background climates as described in Sect. 2.4 and Sect. 2.5. To simulate the Earth's atmosphere, CESM utilizes the Community Atmosphere Model version 5.3 (CAM5, Neale et al., 2012). Here we couple CAM5 to a nudging module to control the atmospheric circulation as described in Sect. 2.2. For the representation of land surface processes, CESM uses the Community Land Model version 4 (CLM4, Lawrence et al., 2011; Oleson et al., 2010). CAM5 and CLM4 are both run on $0.9°$ x $1.25°$ horizontal resolution with 30 layers up to 2 hPa for the atmospheric component (CAM5) and 10 hydrologically-active soil layers down to 3.8 m for the land component (CLM4). Solar forcing follows the model default historical data until the end of 2005 and historical and future simulations of the forcing compiled for CMIP6 thereafter, as in Wehrli et al. (2019, for CMIP6 solar forcing see Matthes et al., 2017). Aerosols and land use/ vegetation follow CMIP5 recommendations for the 20th century simulation until the end of 2005 and the Representative Concentration Pathway 8.5 thereafter (RCP8.5, van Vuuren et al., 2011). Likewise, greenhouse

**Table 1.** Overview of simulations

| Name | Year | Atmospheric forcing | CMIP5 MMM warming [°C] | Actual warming [°C] |
|---|---|---|---|---|
| climatology | 1981–2010 | historical+RCP8.5 | – | – |
| historical | 2018 | historical+RCP8.5 | 1.12 | 1.24 |
| natural | 2018 | pre-industrial+historical | 0.00 | 0.00 |
| warming15 | 2028 | RCP8.5 | 1.50 | 1.60 |
| warming20 | 2042 | RCP8.5 | 2.00 | 2.18 |
| warming30 | 2064 | RCP8.5 | 3.00 | 3.27 |
| warming40 | 2085 | RCP8.5 | 4.00 | 4.39 |

Year corresponds to the year analysed in this study and atmospheric forcing refers to the solar, aerosol, and greenhouse gas forcing. The warming of the CMIP5 multi-model mean (MMM) is given relative to a pre-industrial time period (1861–1880) and corresponds to the target warming level for the respective simulation years. Per design the temperature for the natural simulation is set as reference for no global warming. The actual warming in the CESM simulations differs from the CMIP5 MMM.

gases (GHGs) follow CMIP5 historical recommendations and then after 2005 they are prescribed from observations for $CO_2$, $CH_4$, $N_2O$, and RCP8.5 for other GHGs. Observed global means of $CO_2$, $CH_4$, and $N_2O$ were obtained from NOAA ($CO_2$: Dlugokencky and Tans, 2019; $CH_4$: Dlugokencky, 2019; and $N_2O$: NOAA Earth System Research Laboratory, 2019).

Each experiment is run for four years; thus for the historical simulation the first three years (2015–2017) are used as spin-up, and 2018 is analysed. A climatology for the historical simulations is obtained from a longer simulation that covers the years 1981–2010.

## 2.1 Natural and warming scenarios

The NH2018 event serves as a reference to investigate its characteristics in hypothetical conditions, or storylines, with the same atmospheric circulation but different levels of global warming. In addition to the historical simulation we run one simulation with pre-industrial-like conditions ("natural") and four simulations that follow global warming scenarios. An overview is given in Table 1. For the natural simulation volcanic aerosols and solar radiation are set to historical conditions, whereas we use pre-industrial GHGs, aerosols and SSTs (Sect. 2.4). The four warming scenarios are designed to match 1.5°C, 2°C, 3°C and 4°C global mean warming with respect to pre-industrial conditions (1861–1880) of the CMIP5 multi-model-mean (MMM). We will hereafter refer to these experiments as warming15, warming20, warming30 and warming40. Aerosols, GHGs and SSTs follow RCP8.5. The actual warming of the scenarios slightly differs from the target values, which will be discussed in Sect. 4. All simulations are nudged towards 2015–2018 atmospheric circulation (1981–2010 for the climatology).

## 2.2 Nudging of the atmospheric circulation

To impose the large-scale circulation of the event year (2018) in the model, we use atmospheric nudging of the horizontal winds. The approach is described and validated in Wehrli et al. (2018) and Wehrli et al. (2019). The horizontal winds are relaxed toward observations using a height-dependent nudging function (for the profile see Wehrli et al., 2018). At the surface the nudging strength is set to zero, meaning that the land surface can interact with the atmosphere through surface turbulent fluxes, resulting in balanced surface climate and winds. The large-scale circulation (mostly above 700 hPa) is forced to follow the observations and thus ensures that the observed large-scale weather patterns are reproduced (see Kooperman et al., 2012; Wehrli et al., 2018; Zhang et al., 2014). As a proxy for the observed winds, we use zonal and meridional 6-hourly wind fields from the ERA-Interim reanalysis (Dee et al., 2011). The nudging of the circulation ends on July 27th 2018 due to availability of the input fields.

## 2.3 Determination of warming levels in CMIP5

To produce the ocean fields for the natural and warming scenarios we use model output from the historical and RCP8.5 emissions scenarios from the CMIP5 data archive. We use one simulation per model ("r1i1p1"). To find the years corresponding to the chosen warming scenarios in the CMIP5 ensemble we use near-surface air temperature ("tas"). The pre-industrial reference period is given as 1861–1880. Warming levels with respect to pre-industrial are then determined by taking the difference between annual global mean temperature from RCP8.5 and pre-industrial for each model individually. A 21-year centred running mean is applied to the differences and we then compute the MMM. Following this approach, we obtain a CMIP5 MMM warming of $1.12°C$ for 2018. The $1.5°C$, $2°C$, $3°C$ and $4°C$ MMM warming levels are reached in 2028, 2042, 2064 and 2085, respectively.

## 2.4 Sea surface temperatures representative of prescribed warming levels

To derive SSTs consistent with the different background climates we add delta SST fields to the observed monthly SSTs. The delta SST fields are computed from the CMIP5 SST fields ("tos"), which were regridded to a common grid of $1° \times 1°$. We first apply a 21-year running mean over the monthly merged historical + RCP8.5 tos data and then average over the models. This way, temporally-smoothed monthly fields are created for 2018 (as well as for the three years spin-up: 2015–2017) and the years matching the warming levels: 2028, 2042, 2064 and 2085 (as well as three years spin-up for each). Transient delta SSTs for the warming scenarios are then computed by subtracting the temporally-smoothed fields for the present-day period (2015–2018) from the fields for the warming levels (e.g. 2025–2028 for the $1.5°C$ warming simulation). A monthly climatology is computed over the pre-industrial reference period (1861–1880) and the delta SSTs for the natural simulation are then computed by subtracting the present-day period from this pre-industrial climatology. These delta SSTs for the natural and warming scenarios (see also Fig. 2a) are then added to the historical SSTs of the model to create the SSTs that are prescribed in our scenarios. The detailed step-by-step procedure is given in the Appendix (B1).

## 2.5 Generation and prescription of sea ice

Although it would be possible to derive a delta field for sea ice similarly to the SSTs, this would result in a sea ice field that is not in balance with the new SSTs. Therefore, we derive a relationship between sea ice fraction anomalies and SST anomalies with respect to a climatology, similarly as in the "Half a degree additional warming, prognosis and projected impacts" experiments (HAPPI; Mitchell et al., 2017). Monthly SST and sea ice anomalies are computed for the years 1996–2015 from the climatology of the same years, using ocean observations from NOAA OIv2. A linear regression is then fitted to the

anomalies for each month-of-the-year, longitude and for both hemispheres separately. As an extension of the method applied in the HAPPI experiments we only consider grid cells that undergo a change of sea ice fraction of over $50\%$ for the month in consideration. This ensures that grid cells that are not experiencing enough variability during 1996–2015 are excluded from the analysis. For example, a grid cell close to the North Pole may always have a sea ice fraction larger than $90\%$ while the ocean temperature usually changes only minimally and does therefore not allow for a robust estimation of the slope and intercept

of the relationship. Consequently, the given grid cell would not melt even under very high global warming. To compute the regression of a grid cell we pool SST and sea ice anomalies of all valid grid cells that are within three grid cells to the west and to the east and along the meridians in the same hemisphere. Should no valid grid cells be in this area, more grid cells in the longitudinal direction are included gradually (extending box approach). If a maximum of 11 grid cells to the west and to the east is reached and no valid grid cells are found, a hemispheric linear fit is used. The slope and intercept from this regression

are smoothed zonally using a $500\,\mathrm{km}$ smoother as is done for HAPPI. We further tested an approach where not all grid cells along a meridian were included, but up to two grid cells to the north, south, east, and west of a specific grid cell (5 x 5 grid cells in total). The results are very similar to the first method (not shown). We choose the first method because the resulting sea ice field is overall smoother. The new set of SSTs and sea ice for each scenario is adjusted according to the constraint of Hurrell et al. (2008), which ensures that (i) sea ice fraction is $100\%$ at $-1.8°C$, and SSTs do not get colder than that, (ii) there is no

sea ice at water temperatures warmer than $4.97°C$, and (iii) that within this temperature range the maximum sea ice fraction is limited by a temperature-dependent function.

We contrast our method to the one developed for the "Climate of the 20th Century Plus Detection and Attribution" project (C20C+DA; Stone et al., 2019). For this method, a linear relationship is determined using absolute values of SST and sea ice coverage instead of anomalies. The regression is calculated by pooling all ice-covered grid cells of the Northern and Southern

Hemisphere, respectively. Ice coverage is binned in 100 equally-sized bins and the median SST for each ice bin is determined. The line through the centre of mass of all the bin medians is then estimated by a linear fit. This method was developed to compute sea ice estimates for natural historical simulations (i.e. a cooling). Therefore, it comes with an algorithm that prevents ice from melting and only adds new ice where SSTs cool. As we require a method that works for positive and negative delta SSTs we do not implement the full algorithm but only make use of the linear relationship. For C20C+DA the years 2001–

2010 were used to determine the relationship. For consistency we take the years 1996–2015. Again, we apply the constraint of Hurrell et al. (2008). We hereafter refer to this as the C20C method.

## 2.6 Data sets and data analysis

The atmospheric nudging uses ERA-Interim 6-hourly horizontal wind fields. Mean daily near-surface temperature fields are retrieved from ERA-Interim for comparison to our results. Observed daily maximum near-surface temperatures (TX) are obtained from ERA-Interim and Berkeley Earth (Rohde et al., 2013a, b). Mean daily near-surface temperature, TX and precipitation are retrieved from the Modern-Era Retrospective analysis for Research and Applications, Version 2 (MERRA-2, Gelaro et al., 2017). The data sets are remapped to the resolution of CESM using second-order conservative remapping (Jones, 1999). Results are shown for the absolute values of the variables as well as for anomalies with respect to the 1981–2010 climatology.

We estimate observed regional trends in TX as a function of global mean temperature. Therefore, we use TX from Berkeley Earth and the Climate Research Unit (CRU) data set CRU TSv4.03 (Harris et al., 2020). For global mean temperature GIS-TEMPv4 (GISTEMP Team, 2020; Lenssen et al., 2019) and HadCRUT4 (Morice et al., 2012) are used. The regional averages are computed using the original resolution of CRU TSv4.03 and Berkeley Earth. The regional trends are computed using a linear regression for 1901–2017. Uncertainties of the trend are estimated using the covariance of the residuals of the fit. This gives four trend estimates, whereof only two are shown later on, corresponding to the steepest and flattest estimates for most regions. All possible estimates and their uncertainties are given in Table A1.

## 2.7 Bias correction

Absolute values for TX are bias-corrected using empirical quantile mapping, which corrects the entire distribution of a variable as described in Déqué (2007). We apply the implementation by Rajczak et al. (2016), which is available in the R package qmCH2018 (see also "Code availability"). We choose 1981–2015 as the calibration period and the quantile mapping correction is calibrated for each day-of-the year using a 91-day moving window. The model distribution is translated to the distribution of a reference using 99 quantiles. For model values that are more extreme, quantile mapping uses the correction function of the 99th (1st) quantile to correct values above (below) the calibrated quantiles (Themeßl et al., 2012). We used Berkeley Earth and ERA-Interim as reference for the modeled TX. In the following results for Berkeley Earth are shown but the results for ERA-Interim are very similar (differing by less than 2.5% for the regional averages of the study region).

A mean bias correction specific to the day-of-year was tested but results were found to agree less with the reference data from ERA-Interim and Berkeley Earth (not shown). Note that only absolute TX values are bias-corrected. In cases where anomalies are shown for TX as well as other variables, no correction of the original model output is performed.

## 2.8 Study regions

We show results only north of 25°N. For regional averages we choose the regions defined in the IPCC Special Report on Managing the Risks of Extreme Events and Disasters to Advance Climate Change Adaptation (SREX; Seneviratne et al., 2012). We show results for the European and American regions as well as the Eastern Asia region (EAS) as highlighted in Fig. 1. Note that the southern extent of EAS was cropped such that it only extends from 25°N to 50°N. Ocean grid points are excluded from the analysis. We also focus on a region north of 30°N that is especially vulnerable to extreme conditions because

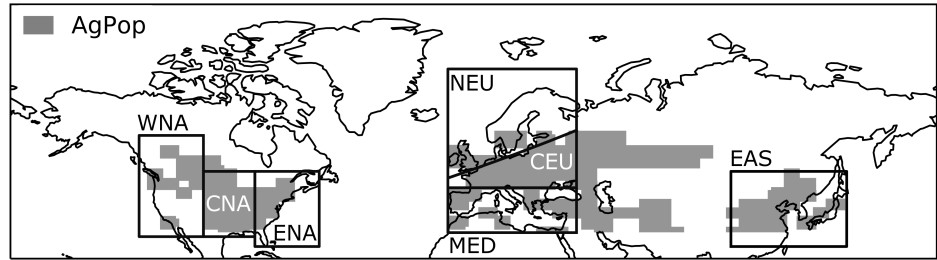

**Figure 1.** Study regions. Shown by the grey shading are regions north of 30°N with high population density and/ or high importance for agriculture (AgPop). The black outlines mark the location of selected SREX regions in America, Europe and Eastern Asia. Note that the Eastern Asia region (EAS) was cropped and extends only from 25°N to 50°N.

it is either densely populated (more than 30 km$^{-2}$) and/ or an important area for agriculture. We define this "human-affected and human-affecting" region following Seneviratne et al. (2018) and Vogel et al. (2019) and refer to it as AgPop (see Fig. 1).

## 3 Results and discussion

We first evaluate the sea ice reconstruction method introduced in this study against observed sea ice and compare the performance to the C20C method in Sect. 3.1. Then we present the results for the natural and warming scenarios in Sect. 3.2 and discuss their implications for possible future events. In Sect. 3.3 we discuss how TX scales with an increase in global mean temperature and if the NH2018 atmospheric circulation influences this relationship.

### 3.1 Evaluation of sea ice reconstruction

Historical sea ice fields are modelled using the methods described in Sect. 2.5 and evaluated against observed sea ice field from NOAA OIv2 (see Appendix Fig. A1). While the C20C method generally overestimates sea ice, over- and underestimation nearly balance out when taking a time average over multiple years for the method presented here (Fig. A1a,c). Hence, the spatial root mean squared error (RSME) for ice grid cells is $< 1\%$ using the new method while it is around $5\%$ using the C20C method. For a single year the errors are larger, $8.5\%$ and $5.6\%$ for the year 2018 for the C20C and the new method, respectively (Fig. A1b,d). It is not possible to evaluate the performance of the sea ice reconstruction for the natural and warming scenarios. The two methods suggest a change of $+10\%$ to $+14\%$ in the annual mean with respect to currently observed sea ice fraction for the natural scenario, $-4\%$ to $-7\%$ for the 2°C warming scenario and $-26\%$ to $-27\%$ for 4°C warming (Fig. 2b,c, see Appendix Fig. A2 for the absolute ice fields for the new method). Note that the two methods show larger differences for the natural scenario but agree better for the warming scenarios.

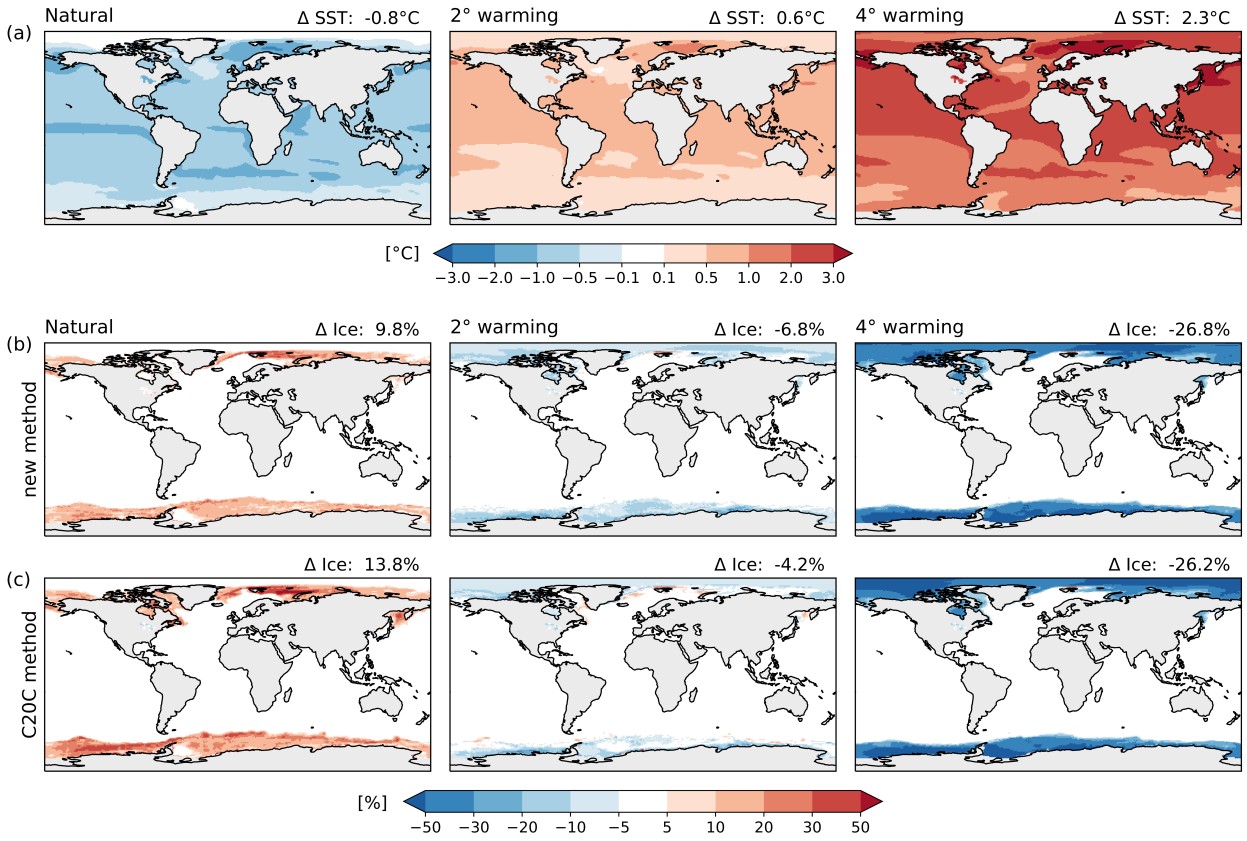

**Figure 2.** Change in SST and sea ice coverage with respect to historical conditions from NOAA OIv2 for the natural (left), warming20 (middle) and warming40 (right) scenarios. (a) Average delta SST for 2018 under the three climate scenarios. (b) Average sea ice coverage change for 2018 under the three climate scenarios using the sea ice estimation method developed in this study. (c) Same as in the (b) but using the C20C method to compute the change.

## 3.2  Storylines for the NH2018 event

During late June and the first weeks of July 2018 an exceptionally strong and persistent Rossby wavenumber 7 pattern domi-
nated the weather in the Northern Hemisphere (Drouard et al., 2019; Kornhuber et al., 2019b). The wave pattern was associated
with almost simultaneous extreme events across the entire hemisphere including heat waves in the western United States, east-
ern Canada, Asia and large areas in Europe (Kornhuber et al., 2019b; NOAA, 2018; Vogel et al., 2019). In Europe, the July
temperature anomaly was ranked second highest on record, just one hundredth degree Celsius behind 2015 (NOAA, 2018).
Vogel et al. (2019) show that the area of important "human-affected and human-affecting" regions (AgPop) experiencing si-
multaneous heat waves peaked at the end of July. Hence, we focus here on the month of July 2018 and especially on a 15-day
period at the end of the month from July 13 to 27 (which also corresponds to the last days of our simulations where the at-

mospheric nudging is available). Note that strong heat waves also occurred after July 27, which were locally more extreme than the heat waves examined during our study period. For example in Spain and Portugal the heat wave peaked during the first week of August (Barriopedro et al., 2020), the Netherlands experienced a second heat wave starting at the end of July and lasting for the first week of August (KNMI, 2020), and South Korea reported new record daily maximum temperatures in the beginning of August (KMA, 2019).

A comparison of the temperature anomaly to ERA-Interim shows that the daily and 15-day mean anomalies are well represented in the nudged historical CESM simulation (Fig. 3). Similar results are obtained for MERRA-2 (not shown). For 13 to 27 July 2018 the mean daily temperature in the historical simulation is on average $5°C$ warmer than the climatology (1981–2010) for large areas in Scandinavia and some smaller areas in Germany, Belgium and the Netherlands (Fig. 3i). In Northern America temperatures are $2.5°C$ to $5°C$ above average for Newfoundland, Québec, Texas and northern Mexico as well as large parts of the Western North American (WNA) and EAS regions. An intensification of the hot anomaly can be seen for the Central European (CEU) and Northern European (NEU) regions during July (Fig. 3d,e), whereas in the AgPop region the anomaly is around 1.5 to $2°C$ for the entire month (Fig. 3h). TX over the same time period is simulated even more than $7.5°C$ warmer than the climatology in much of Scandinavia and Northern Germany, Belgium and the Netherlands (Fig. 4b and Appendix Fig. A3a-d for comparison to ERA-Interim, MERRA-2 and Berkeley). In contrast, TX for much of the Mediterranean (MED) region is close to the climatological average. Colder-than-average TX is seen for areas surrounding the Aegean Sea and Black Sea, as well as for Portugal, parts of Spain and for the United States east coast (Fig. 4b). Note that in Portugal and Spain a heat wave developed in the first week of August (Barriopedro et al., 2020), which is, however, outside our study period. The temporal evolution of the TX anomaly during the month of July resembles that of the daily mean temperature for the study regions (Appendix Fig. A4).

In the nudged historical simulation maximum daytime temperatures (TXx) exceeding $40°C$ are simulated for parts of the Central North American (CNA) and WNA regions and east of the Caspian Sea. The fraction of the AgPop region affected by TXx $> 40°C$ is 9% (Fig. 4a). In the natural scenario only 7% of the AgPop region is affected by such high temperatures. On the other hand, the warming scenarios show a strong intensification of the magnitude and extent of the event. In the warming20 scenario the fraction of the AgPop region experiencing $> 40°C$ temperatures increases to 13% of the total area and more than one third (34%) is affected in the warming40 scenario. At $4°C$ global warming the model predicts that most of the United States will experience temperatures above $> 40°C$ given the 2018 circulation pattern (Fig. 4a). For much of CNA anomalies with respect to 1981–2010 exceed $10°C$ (Fig. 4b). Large areas of CEU and NEU experience temperatures around $38°C$ and higher, which corresponds to more than $10°C$ above climatology for Scandinavia.

The NH2018 event did not only bring exceptionally warm temperatures to central and north-western Europe but also a dry spring and summer contributed to severe drought conditions (Toreti et al., 2019). In contrast, south-eastern Europe experienced a wetter-than-usual spring and summer (Toreti et al., 2019). This marked precipitation dipole over Europe is reproduced in the historical simulation (Fig. 5). In the warming40 scenario we find a decrease of up to $-60\%$ in precipitation over Mediterranean Europe compared to the historical simulation (Fig. 6a). These changes in precipitation counteract the precipitation dipole observed in the historical simulations but precipitation still remains above climatology for most of Mediterranean Europe (not

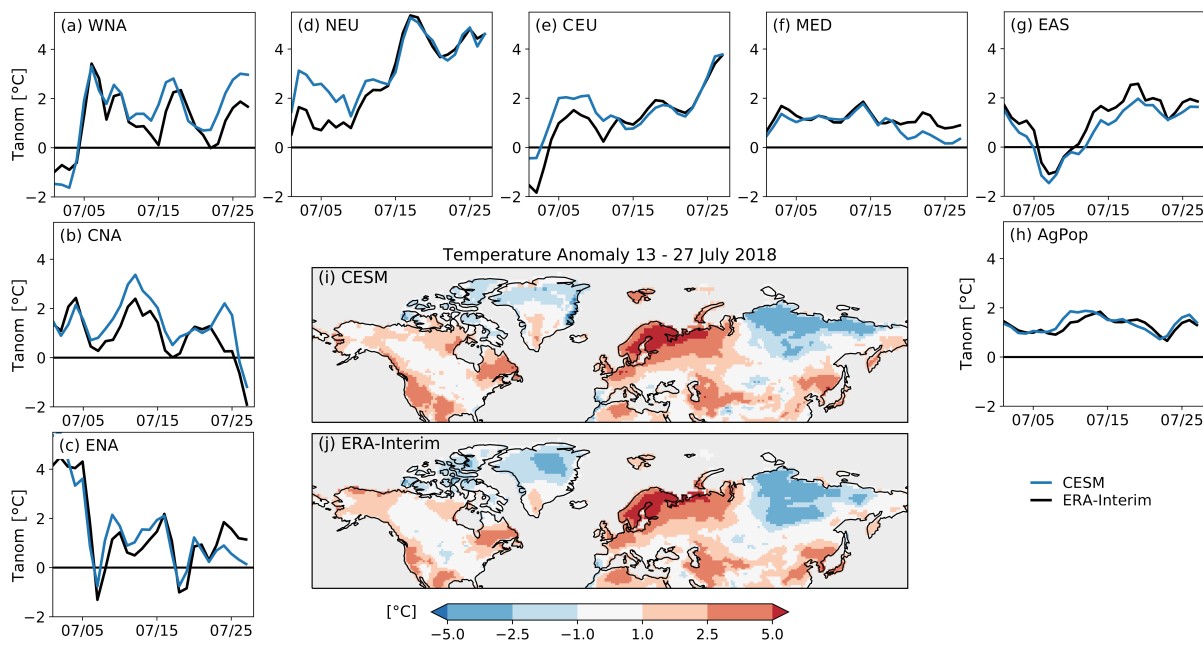

**Figure 3.** Anomalies of daily mean temperature with respect to the 1981–2010 climatology. The maps show the average anomaly over 13 to 27 July 2018 for (i) the nudged historical CESM simulation and (j) for ERA-Interim. The time series show the evolution of daily temperature anomaly (Tanom) for CESM (blue) and ERA-Interim (black) for the month of July averaged over (a-g) seven SREX regions as well as (h) for the AgPop region.

shown). Additionally, the warming scenarios simulate higher net surface shortwave radiation (Fig. 6b) in large parts of Europe resulting in an increase in sensible heat flux and a decrease in latent heat flux (not shown). In Eastern Asia very dry conditions are observed and simulated for Japan and the Korean Peninsula, whereas rather more than climatological precipitation can be seen for the neighbouring regions in China (Fig. 5). No substantial precipitation decrease is simulated for the warming scenarios

(Fig. 6a) although total cloud cover fraction slightly decreases (not shown). Higher net shortwave radiation (Fig. 6b) leads to an increase in sensible heat flux in the warming40 scenario, especially over Japan and Korea (not shown). Over North America a contrasting precipitation pattern is simulated with a strong positive anomaly in the east and a precipitation deficit in the west (Fig. 5a). Over a smaller region in CNA there is a precipitation decrease of around $-40\%$ in the warming20 scenario and up to $-60\%$ in the warming40 scenario. This is in contrast to observed summer precipitation trends showing an increase in the

central United States over the last century (Alter et al., 2018; Wuebbles et al., 2017). The precipitation decrease co-occurs with a decrease of up to $25\%$ in total cloud cover fraction for central North America in the warming40 scenario (not shown) as well as higher net shortwave radiation at the surface for large parts of North America (Fig. 6b). Further, the increase in net shortwave radiation goes along with a decrease in latent heat flux and an increase in sensible heat flux, which is most pronounced in the warming40 scenario for CNA (not shown). This change in the surface fluxes implies a reduction in evaporative cooling and

increase of near-surface heating, which can amplify the heat wave (Fischer et al., 2007; Seneviratne et al., 2006, 2010). Hence,

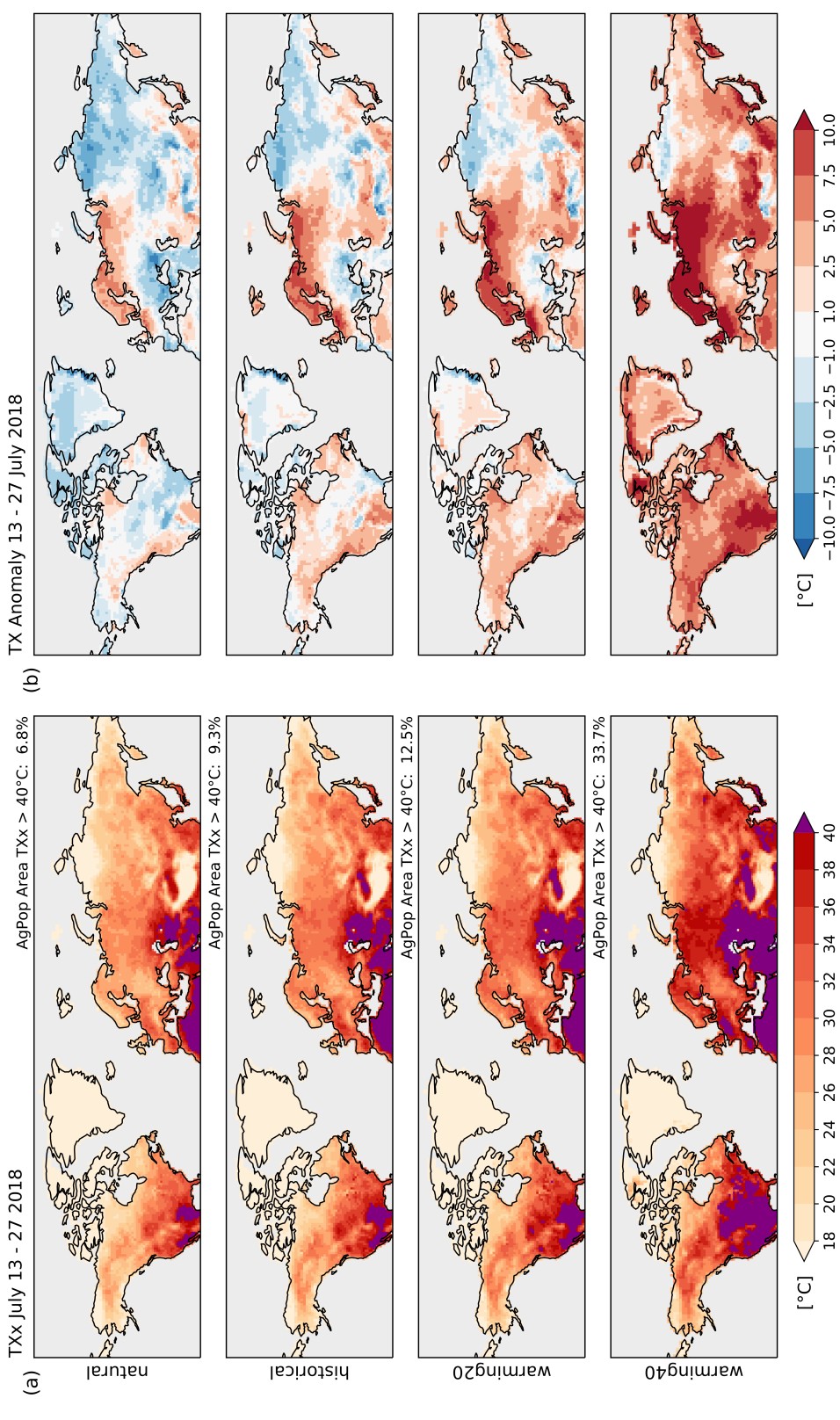

**Figure 4.** Daily maximum temperatures (TX) for July 13 to 27 2018. (a) Bias corrected absolute maximum TX (TXx) over the 15-day period. The number in the upper right corner indicates the fraction of the AgPop region where absolute temperatures exceed $40°C$. (b) Anomaly of mean TX over the 15-day period compared to the 1981−2010 historical climatology (no bias correction).

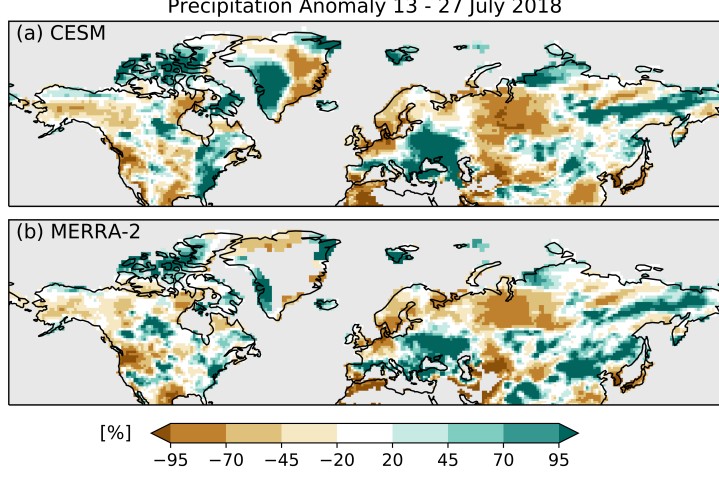

**Figure 5.** Precipitation anomaly (%) for July 13 to 27 2018 compared to the 1981–2010 climatology. (a) CESM historical simulation (b) MERRA-2. Land areas with less than 0.1mm/day precipitation in the climatology are masked out.

in a warmer climate the heat waves during a NH2018-like event might be amplified through land-surface feedback. Further, some of the regions experiencing precipitation excess might be seeing less precipitation, for example the Mediterranean. This agrees with findings by Toreti et al. (2019) showing that the projected likelihood of anomalously wet conditions as observed during NH2018 decreases for southern Europe. On the other hand, in the natural scenario the event is less extreme due to

higher total cloud cover (not shown), less net shortwave radiation (Fig. 6b), an increase of precipitation, especially for CNA (Fig. 6a), and higher soil moisture (not shown). Together with the colder background climate these effects combine to reduce the maximum reached temperatures and abate the heat stress for example for crops. In short, we find that the NH2018 event would have been less widespread and less hot under natural climate conditions. Contrarily, it would have affected an even much larger area and would have caused particularly dangerously high temperatures and severe drought conditions in a large fraction

of the AgPop region under higher levels of global warming.

### 3.3   Scaling of local temperature increase with global warming

We scale the increase in mean TX for 13–27 July of each study region with the global mean temperature change in our scenarios. As our simulations are all nudged toward the 2018 atmospheric circulation, we compare the results to simulations for July mean TX with random circulation from CMIP5 and in specific to the CESM simulations in CMIP5 (CMIP5-CESM). This way it is

possible to disentangle the effect of the specific circulation pattern from the global mean warming trend qualitatively (note, however, that the CMIP5 models including CMIP5-CESM use an interactive ocean). The simulation years from the CMIP5 models are chosen to equal the actual warming for the CESM simulations (Table 1). In general, the increase in TX with global mean warming between $1.2°C$ and $4.4°C$ follows a linear relationship (Fig. 7). For MED both CESM simulations as well as the CMIP5 MMM lie close together and there is little spread between the CMIP5 models (Fig. 7f, orange shading). This indicates

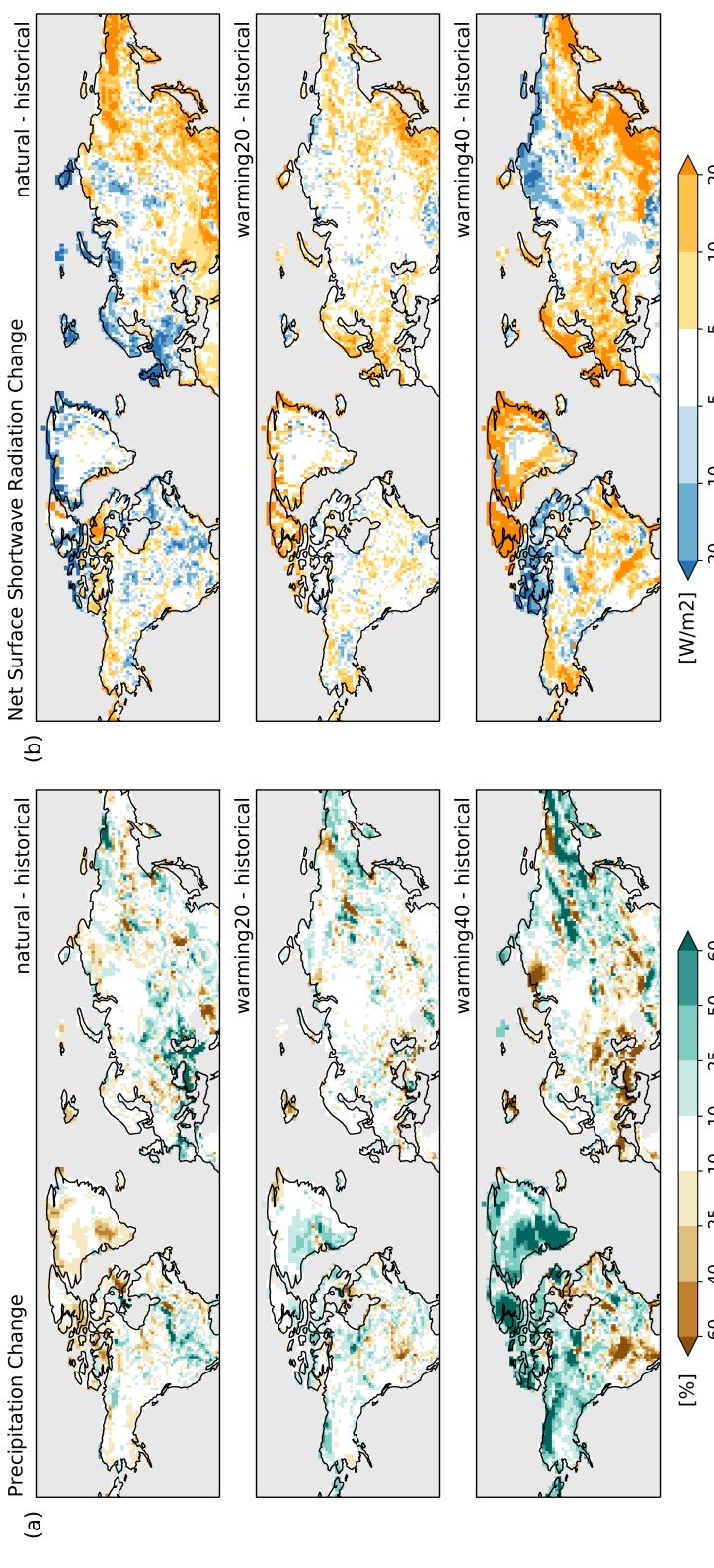

**Figure 6.** (a) Changes in precipitation (%); and (b) net shortwave radiation at the surface (W m$^{-2}$). Changes are averaged over 13 to 27 July 2018 and are given as anomalies from the historical simulation. Land areas with less than 0.1 mm/day precipitation in the historical simulation are masked out in (a).

that the CESM model behaves similarly to the CMIP5 MMM and that there is no change in the relationship induced by the atmospheric circulation of 2018. Hence, the increase in TX is driven by the background global warming. For WNA and CEU both CESM model configurations simulate a higher TX for a given warming level than the CMIP5 MMM (Fig. 7a,e), indicating that CESM produces warmer future climate in general for these regions. However, the nudged CESM simulations show a very similar increase of TX as CMIP5-CESM, which implies that for these regional averages the 2018 circulation did not alter the

relationship to the global mean temperature. Both SREX regions cover a rather large and climatologically diverse area. In the nudged historical simulation positive TX anomalies are observed in the western part of the CEU region and negative anomalies in the east (Fig. 4b). The two circulation-related contributions likely compensate each other when computing the regional average for the historical simulation and also for the warming scenarios, which might be the reason why the scaling is similar to the non-nudged simulations. For EAS the nudged CESM and CMIP5-CESM show higher TX at higher global warming

levels than the CMIP5 MMM but not at lower global warming levels (Fig. 7g). The slope of the nudged CESM is steeper than of CMIP5-CESM, indicating a contribution by the atmospheric circulation pattern of 2018. For NEU, CNA and Eastern North America (ENA) TX from the nudged CESM simulations shows a steeper increase with global warming compared to CMIP5 and CMIP5-CESM (Fig. 7d,b,c), whereas the latter two behave similarly. At the highest level of global warming (4.39°C) TX from the nudged simulations even exceeds the envelope of CMIP5 models. Hence, there is an effect of the circulation pattern

in these regions which linearly increases for the same event at higher global warming levels. For ENA TX is mostly around climatological values or even below in the historical simulation. Therefore, it is counter-intuitive that for the warming scenarios TX increases due to the same circulation pattern. We speculate that this is related to the increase in shortwave radiation seen for ENA at higher global warming scenarios (Fig. 6b). For CNA there is a steeper increase of TX between the 2.18°C and 3.27°C global mean warming which might be related to a change in surface heat fluxes and stronger land-atmosphere coupling as was

found in Sect. 3.2. The AgPop region spans across several of the SREX regions evaluated here, mainly EAS, CNA, CEU and MED. Therefore, the relationship is a combination of the effects described above. For the relationship between mean daily temperature in July and global warming the results look similar except for the EAS and AgPop regions where the differences between the nudged CESM and CMIP5-CESM disappear (not shown).

     We compare the simulated trends to the observed trends over the last century (extrapolated to higher global warming levels).

For North America, especially CNA (Fig. 7b) and ENA (Fig. 7c), the regional warming is stronger in the CMIP5 models compared to observations. It has been documented in previous articles that temperature trends are commonly overestimated by CMIP5 models for North America (e.g. Alter et al., 2018; Donat et al., 2017), which was shown to be due to an overestimation of soil moisture-temperature coupling (Sippel et al., 2017; Vogel et al., 2018) and an increase of irrigation that is not included in the models (Alter et al., 2018; Thiery et al., 2020). Warming seems to be overestimated for the same reasons in EAS (Fig. 7g,

see also Donat et al., 2017). In the latest CMIP6 ensemble, these biases appear smaller (Seneviratne and Hauser, 2020). Depending on the observational data sets used, there can be large differences in the observed trend for some of the regions (e.g. CEU and MED: Fig.7e,f). The trend estimates from the four possible combinations of observational data for global mean temperature and TX – including the uncertainty of the fit of the linear regression – are given in Table A1. Observational uncertainty can substantially affect the trend estimate for example due to incomplete coverage and data infilling (Cowtan

and Way, 2014); different data quality control and bias correction methodologies (Morice et al., 2012); as well as due to the inclusion and weighting of different observational data and data types. In addition, both global temperature data sets (HadCRUT4 and GISTEMPv4) blend near-surface temperatures over land with SSTs over the ocean, which differs from the procedure for models where near-surface temperatures are averaged over land and ocean (e.g. Cowtan et al., 2015). Hence, models and observations show different global mean warming rates and it was found that the rate is slower for observations due to the blending of SSTs and near-surface land temperatures, as well as due to very sparse observations of the warming in polar regions (about $1.1°C$ warming in the models since pre-industrial vs. $1.0°C$ in observations; see also Cowtan et al., 2015; Richardson et al., 2016). Therefore, the observed trend shown in Fig. 7 would be smaller if computed in the same way as for the models.

## 4   Conclusions

We present an analysis of scenario storylines building on the extreme 2018 Northern Hemisphere summer ("NH2018 event"). These storylines retell the NH2018 event in alternative worlds with the same atmospheric circulation as observed but different background global mean warming. The event is alternatively simulated in a natural setting without human imprint on the climate system ("natural"), for the present-day climate conditions, and for four scenarios at different levels of global warming ($1.5°C$, $2°C$, $3°C$, and $4°C$). All simulations nudge the large-scale atmospheric circulation toward the 2018 conditions but differ in their greenhouse gas and aerosol forcing as well as in the SSTs and sea ice coverage. The focus of this study is on a period from 13–27 July 2018, when the heat wave affected a large fraction of the populated area of the Northern Hemisphere. It has to be noted, that locally severe heat waves were observed during other time periods, mainly in the beginning of August as for example in Korea and for the Iberian Peninsula (e.g. Barriopedro et al., 2020; KMA, 2019). However, due to the availability of input fields for atmospheric nudging, these later events are not analysed. It is likely that similar effects to those described in this study could be found for other regions, when choosing a time period matching the peak of the heat wave. For these more local features of the NH2018 event a regional climate modeling approach might shed more light on the characteristics of future events and would complement the findings of this study.

The storylines for the NH2018 event show drastic consequences for the entire Northern Hemisphere in case of a re-occurrence of this atmospheric pattern at higher global warming. Maximum temperatures increasingly surpass $40°C$ with large parts of the southern United States experiencing such extreme temperatures already at $2°C$ global warming. At $4°C$ almost the entire United States as well as regions in Western Europe and Eastern Asia are affected. The total area of important "human-affecting and human-affected" regions (Seneviratne et al., 2018; Vogel et al., 2019) in the Northern Hemisphere (north of $30°N$) experiencing temperatures higher than $40°C$ increases from $9\%$ during the NH2018 event to $13\%$ and $34\%$ at a global warming of $2°C$ and $4°C$, respectively. In the "natural" simulation, the fraction of the area affected reduces to $7\%$. It has to be noted that these values are sensitive to the bias correction method and the reference data set chosen for the calibration of the correction. A quantile mapping method was chosen because it agrees well with observed TX during the heat wave but its validity for the warming (and natural) storylines cannot be tested.

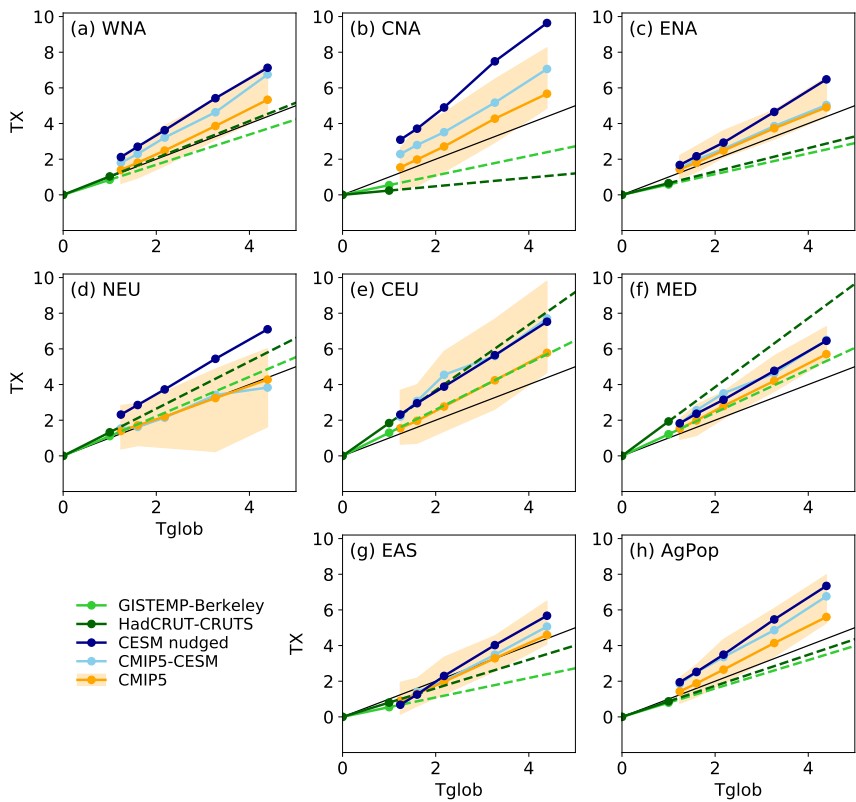

**Figure 7.** Scaling of daily maximum temperature (TX) for the study regions with global mean warming (Tglob). Shown is the mean TX for 13–27 July from the nudged CESM simulations (dark blue), the July mean TX for the CMIP5 multi-model-mean (orange line) and the full model range (orange shading) as well as the July mean TX for CESM from the CMIP5 ensemble (CMIP5-CESM, light blue). The solid green lines correspond to the observed warming while the dashed green lines indicate the extrapolation beyond the observed warming (light green for Berkeley Earth using GISTEMPv4 as reference for global mean temperature change and dark green for CRU TSv4.03 using global mean temperature anomalies from HadCRUT4). The black line indicates the 1:1 line.

We find that TX for the different scenarios increases linearly with global mean warming. For the CNA, NEU and ENA region (and less strongly also for EAS) we find a steeper slope of the relationship given the 2018 atmospheric circulation conditions,

indicating that these regions are affected by an amplification of the heat wave in a warming climate. Comparison with trends from observations indicates that the regional warming trend might be overestimated in the models, especially for CNA and ENA. Nevertheless, the trends agree well for the other regions and it has to be noted that the observed trends also have to be evaluated with caution, due to the uncertainty of the trend fit, observational uncertainty (e.g. Cowtan and Way, 2014; Morice et al., 2012) and due to a different methodology to compute global mean temperatures in observations compared to models

(Cowtan et al., 2015). Further, it is important to keep in mind that even if the increase in temperature is linear, the associated impacts would likely not be. Human well-being, crop yields and fire risk for example are related to certain temperature ranges and effects of the heat wave might be amplified once certain thresholds are surpassed.

It is intrinsic to the applied type of experiment (forced circulation) that there is no atmospheric variability among ensemble members, which prevents an assessment of the probability of the scenarios. The study is thus not designed to answer the

question of how probable it is for a NH2018-like event to re-occur at a certain warming level. Statistically the probability that exactly the same circulation with the same history and evolution during the heat wave will re-occur is small. Further, the nudging approach does not ensure that the circulation pattern is physically in balance with the scenarios for higher global warming or natural conditions. Hence, it might be that it is unlikely that the atmospheric circulation patterns associated with NH2018 event could establish in a warmer climate. However, it has been shown from observations that the occurrence of

the driving NH2018 atmospheric circulation pattern – a stationary wavenumber 7 Rossby wave – has increased significantly in recent years, a possible consequence of the enhanced land-ocean temperature contrast due to global warming (Kornhuber et al., 2019b). Even if there should be no trend in amplitude or persistence of these wave events, associated heat waves in a future climate will be amplified by global warming (Kornhuber et al., 2019a). From a dynamical perspective, it appears thus probable that similar wave events will occur in a warmer climate and thus the study of such hypothetical events is highly relevant.

The global warming levels for the nudged simulations and corresponding atmospheric forcing were chosen based on the global mean warming using near-surface air temperature of the CMIP5 MMM and from this delta SSTs were computed. However, this does not imply that the same warming levels will be reached in CESM. Indeed, the global mean warming in the simulations is higher. It was shown that computing blended global mean temperature from near-surface air temperature and SSTs together with accounting for incomplete data from observations leads to approximately $0.2°C$ less global warming since

the 19th century (Cowtan et al., 2015; Richardson et al., 2016). Applying this to our simulations, the discrepancy between the target warming level and actual warming is reduced. Still, to simulate the target warming with better accuracy a solution could be to choose the years corresponding to a target warming level from CESM itself (using blended global mean temperatures) and also computing the delta SSTs only from CESM (by using the CESM model data from CMIP5 or by running a long, non-nudged simulation with prescribed ocean following e.g. the RCP8.5 scenario). We argue that even if the target warming

might be simulated with better accuracy, the results would not differ fundamentally from those found here as we show that local temperatures in the simulations scale approximately linearly with global warming.

To our knowledge this is the first study to use storylines of different warming levels for a specific event in a global climate model setup. Kornhuber et al. (2019a) show that in the future the wavenumber 7 circulation pattern can lead to major risks in breadbasket regions that are important for crop production. The storyline approach presented here provides insightful results
that help understand the risks and consequences of similar events in a future climate. Our results highlight that large areas of the Northern Hemisphere will suffer from major heat stress given the same circulation at higher background warming levels, which can have dangerous consequences for agriculture, ecosystems, the economy and also human health.

*Code and data availability.* The python code for the sea ice algorithm developed in this study is available upon request. For the quantile mapping we used the R package qmCH2018 version 1.0.1 (https://github.com/SvenKotlarski/qmCH2018; Rajczak et al., 2016), which was
run in R version 3.3.2 (https://www.r-project.org/). CMIP5 data is available from the the Earth System Grid Federation (ESGF).

## Appendix A:  Additional figures and table

This section provides additional figures and tables accompanying the main article.

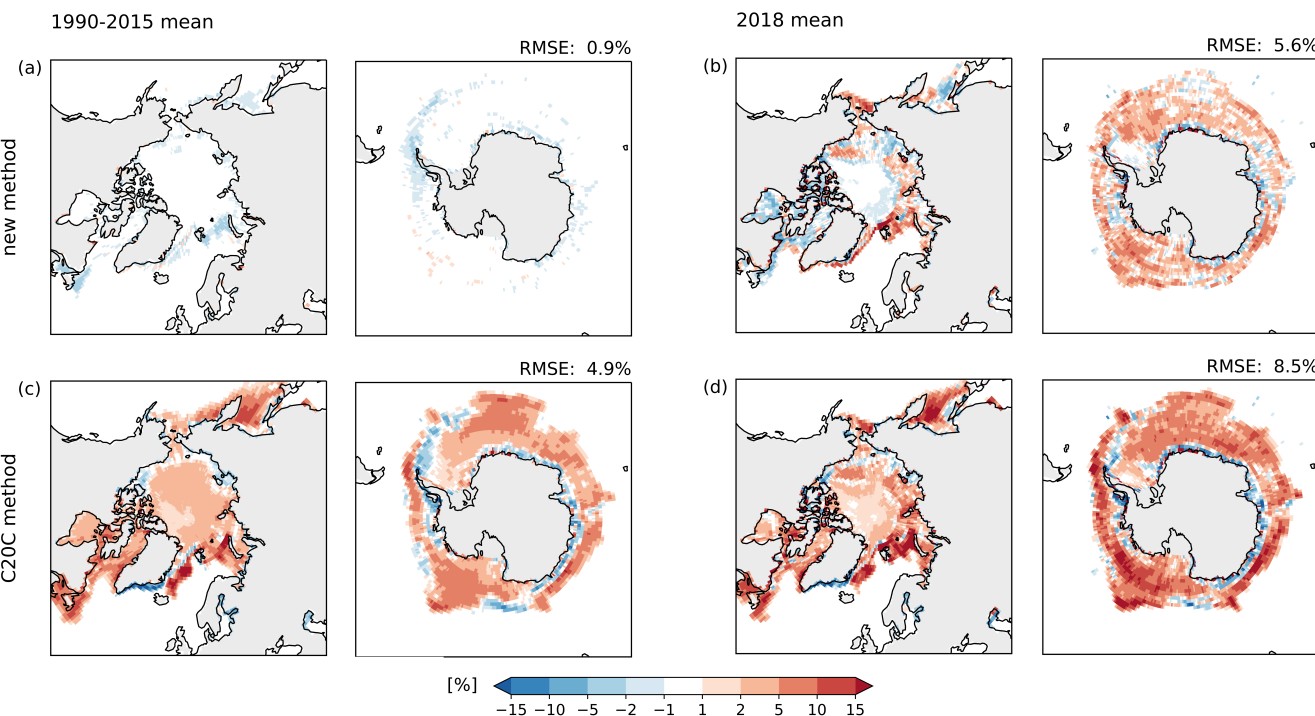

**Figure A1.** Evaluation of sea ice reconstruction. Shown is the bias (%) of reconstructed historical sea ice for the (a,c) 1990–2015 mean and (b,d) 2018 mean compared to NOAA OIv2 for areas north of 50°N and south of 50°S. The new method developed in this study (a,b) is compared to the C20C method (c,d). Numbers in the upper right corner indicate the global mean RMSE for grid cells that are covered by ice in the observed or estimated fields.

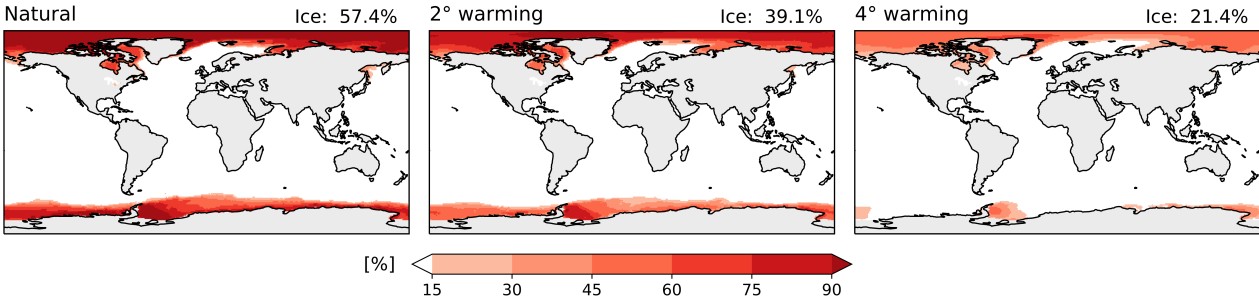

**Figure A2.** Absolute sea ice coverage fractions for 2018 in the natural, warming20 and warming40 simulation determined with the algorithm presented in this study. The numbers in the upper right corner indicate the yearly average sea ice coverage fraction for ice grid cells.

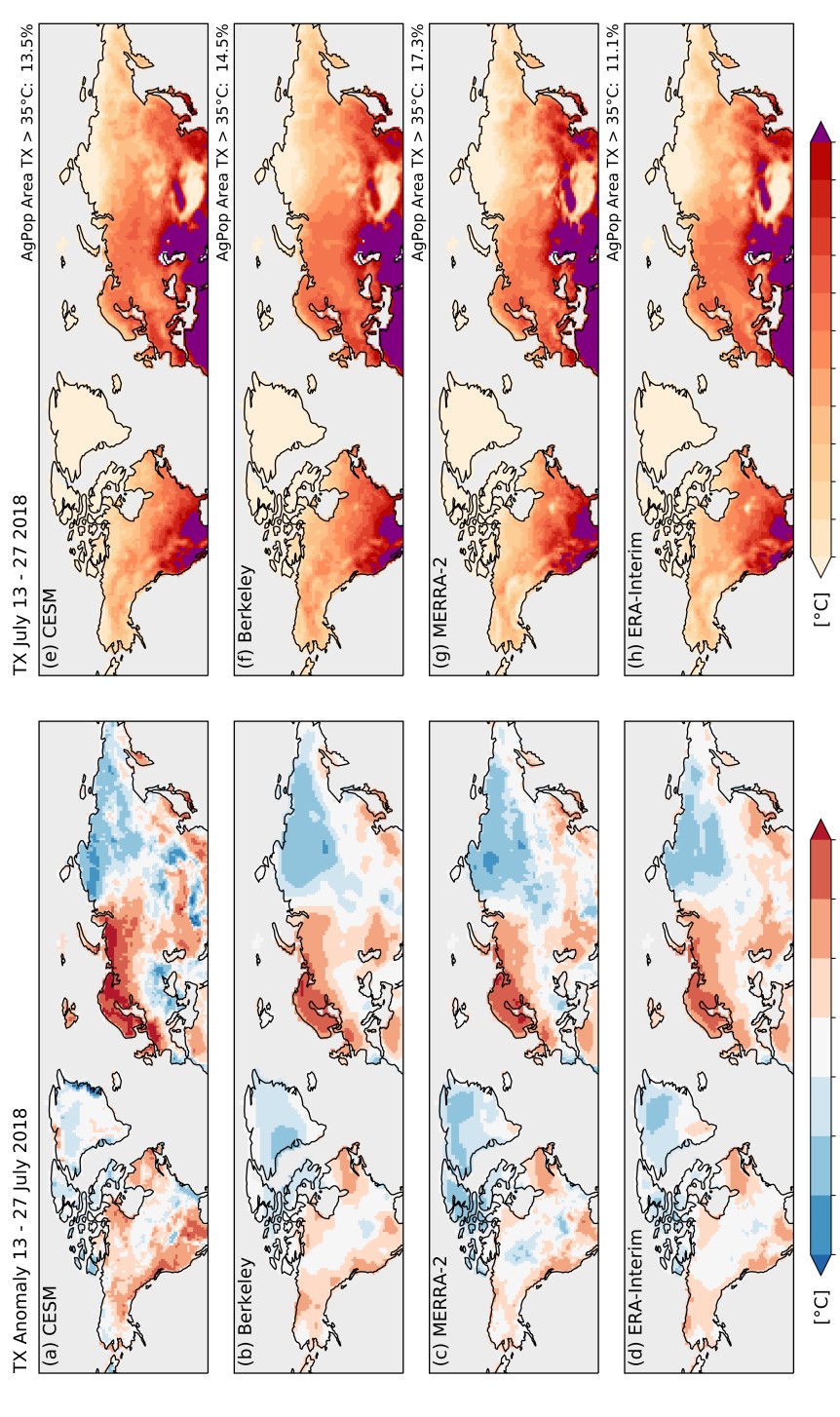

**Figure A3.** Maximum daily temperature (TX) and anomalies of TX averaged over July 13–27 2018. (a) TX Anomalies are shown with respect to the 1981–2010 climatology for the historical simulations; (b) same for ERA-Interim; (c) MERRA-2; and (d) Berkeley. (e) Absolute TX values for CESM are bias-corrected using quantile mapping and Berkeley Earth as reference to calibrate the correction. (f) Absolute TX values from Berkeley; (g) MERRA-2; and (h) ERA-Interim are shown as comparison. In the right corner above e-h the percentage of the AgPop region where TX averaged over July 13−27 2018 exceeds 35°C is given.

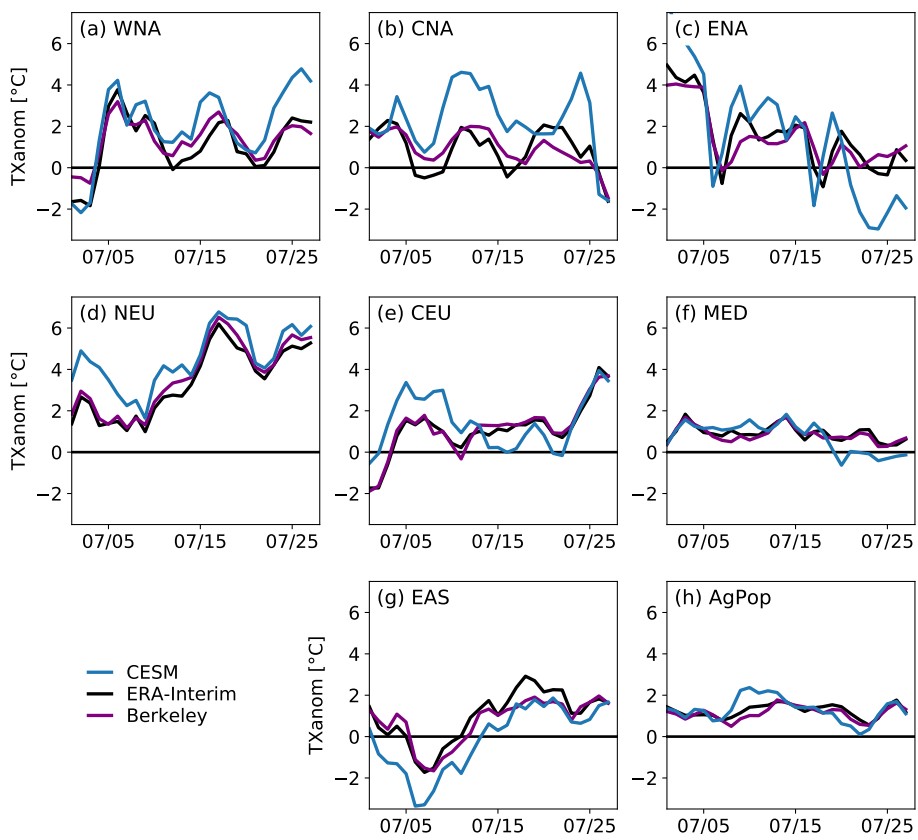

**Figure A4.** Time series showing the anomaly of maximum daily temperature (TXanom) for July 2018 averaged for (a-g) seven SREX regions and (h) the AgPop region. Shown are the historical simulation from CESM (blue), ERA-Interim (black) and Berkeley (violet). The reference climatology is 1981–2010.

**Table A1.** Slopes for all possible combinations of the observational data sets used for the linear regression as well as their uncertainties (one standard deviation).

| TXreg | CRU TSv4.03 | | Berkeley Earth | |
| Tglob | GISTEMPv4 | HadCRUT4 | GISTEMPv4 | HadCRUT4 |
|---|---|---|---|---|
| WNA | $0.85 \pm 0.21$ | $1.03 \pm 0.24$ | $0.85 \pm 0.22$ | $1.04 \pm 0.25$ |
| CNA | $0.16 \pm 0.30$ | $0.24 \pm 0.34$ | $0.55 \pm 0.31$ | $0.68 \pm 0.35$ |
| ENA | $0.56 \pm 0.22$ | $0.65 \pm 0.25$ | $0.58 \pm 0.21$ | $0.68 \pm 0.23$ |
| CEU | $1.52 \pm 0.30$ | $1.84 \pm 0.33$ | $1.30 \pm 0.30$ | $1.60 \pm 0.34$ |
| NEU | $1.13 \pm 0.31$ | $1.32 \pm 0.35$ | $1.11 \pm 0.29$ | $1.29 \pm 0.33$ |
| MED | $1.67 \pm 0.15$ | $1.93 \pm 0.16$ | $1.21 \pm 0.16$ | $1.40 \pm 0.18$ |
| EAS | $0.61 \pm 0.16$ | $0.80 \pm 0.18$ | $0.55 \pm 0.19$ | $0.75 \pm 0.21$ |
| AgPop | $0.72 \pm 0.12$ | $0.87 \pm 0.14$ | $0.80 \pm 0.13$ | $0.96 \pm 0.14$ |

## Appendix B: Ocean forcing files

This section explains in detail how the ocean forcing files are prepared using model output for the historical and RCP8.5 scenario from the CMIP5 data archive.

### B1 Step-by-step generation of delta SST and SST input files

Warming levels are determined using near-surface air temperature ("tas") from CMIP5. Delta SSTs are computed from sea surface temperature fields ("tos").

1. For easier computation later-on and to prevent steep gradients at the continent's edges, the "tos" fields are grid-filled using Poisson's equation relaxation scheme, where this has not been done by the model. All data are regridded to the 1x1 degree grid used in the NOAA OIv2 (Reynolds et al., 2002) and HadISST1 (Rayner et al., 2003) observational products, which are used to prescribe sea surface temperatures and sea ice coverage in CESM.

2. To find the years corresponding to the warming levels weighted global yearly means of "tas" are computed for the historical and RCP8.5 time period. Only models that provide complete monthly data for both "tas" and "tos" for the historical (starting latest 1861) and the RCP8.5 time period (at least until 2099) are included.

3. The pre-industrial reference period is defined from 1861–1880. "Tas" for each model is averaged over this period separately. The warming for each model is computed by taking the difference between the yearly averages from step 2 and the pre-industrial reference period. A 21-year running mean is taken over the yearly warming values first, and then the multi-model mean (MMM) is computed.

4. The first year where the multi model warming exceeds 1.5°C, 2°C, 3°C and 4°C is chosen to compute the delta SST fields in the following and also to set the forcing level for aerosols and GHGs from RCP8.5 for the warming simulations. The numbers in this study are: for the current warming (2018) a value of 1.12° and the 1.5°, 2°, 3° and 4° warming levels are reached in 2028, 2042, 2064, 2085 respectively, for the MMM.

5. Now the delta SSTs are computed. First, a 21-year boxcar filter (running mean over months) is applied to the monthly "tos" fields for the merged historical (1975–2005) + RCP8.5 time period. These ocean fields are saved together with the multi-year monthly averaged fields for the pre-industrial time period (1861–1880).

6. The MMM is computed for the monthly ocean fields. Then the delta SST for the natural scenario is computed by subtracting the MMM pre-industrial monthly climatology from the present-day monthly fields (2015–2018 to include spin-up). The delta SSTs for the warming scenarios are computed by subtracting the present-day monthly fields from the MMM field of the years of the warming scenario under consideration (e.g. for 2° of global warming: 2042 plus the years 2039–2041 for spin-up), hence:

   $deltaSST\_natural = present - pre\text{-}industrial$; for *present* between 2015–2018 and *pre-industrial* averaged over 1861–1880.

$deltaSST\_warming20 = warming20 - present$; for *warming20* corresponding to years 2039–2042 (with the 21-year boxcar filter applied first) and *present* between 2015–2018.

Note that these delta SSTs are transient.

7. We compute the SST input for the simulations by subtracting the natural deltaSST field and adding the warming deltaSST fields to the historical SSTs of the model. The constraint of Hurrell et al. (2008) is used to ensure that temperature is not below $-1.8°$C.

*Author contributions.* KW, MH and SIS designed the experiments and discussed the results. KW developed the sea ice reconstruction method with input by MH and based on code used for HAPPI that was made available by Eunice Lo from University of Bristol. KW ran the model simulation and analysed the results. KW prepared the manuscript with contributions from all co-authors

*Competing interests.* The authors declare that they have no conflict of interest.

*Acknowledgements.* We thank Eunice Lo for sharing the HAPPI sea ice code. We also thank Urs Beyerle and Richard Wartenburger for retrieving the CMIP5 data. We are grateful to Geert Jan van Oldenborgh and an anonymous reviewer for evaluating our manuscript and for their helpful comments. We acknowledge financial support from the European Research Council (ERC) "DROUGHT-HEAT" project (Grant 617518) funded by the European Community's Seventh Framework Programme. KW thanks Sven Kotlarski for his input on bias correction and quantile mapping.

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
