# Peer review of "Storylines of the 2018 Northern Hemisphere heat wave at pre-industrial and higher global warming levels"

_Earth System Dynamics, 2019_

## Referee Comment (RC1) · Anonymous Referee #1 · 24 Mar 2020

The concept of asking how a given meteorological event might have been exacerbated by global warming has been growing in popularity as an alternative to the more common approach of probabilistic event attribution. As first proposed by Trenberth et al. (2015 doi: 10.1038/NCLIMATE2657) and Shepherd (2016 doi: 10.1007/s40641-016-0033-y), this 'storyline' approach takes the atmospheric flow configuration leading to the event as given, and quantifies the impact of global warming conditional on that flow configuration. The arguments for why this may be useful are given in those two papers, but the general concept is that generality is sacrificed in order to obtain a more detailed and hopefully more informative statement of impact (since tied to a particular event). The storyline approach was first applied to synoptic-timescale weather phenomena,

e.g. tropical cyclones, where the conditionality was applied either through the initial conditions in a forecast, or through the boundary conditions in a regional model. Here the storyline approach is applied to a multi-week heat wave event, nudging the circulation in a global model to reanalysis, following the methodology previously used by the authors in their 2019 JGR paper to understand the role of soil-moisture feedbacks in heat waves.

It is important to document applications of the storyline approach in different contexts so that we can learn to understand its strengths and weaknesses. From that perspective this study is welcome, and for the most part the results are carefully explained and clearly presented. I find Figure 7 to be the most interesting of all. I am happy to recommend publication, provided the following points are addressed:

1. Figure 3 shows only temperature anomalies. It would be good to also show absolute temperatures (e.g. in the maps), so that the reader can see the extent of the temperature bias of the model.

2. In some periods and regions, the differences between the nudged run and ERA-Interim anomalies in the time series in Figure 3 can exceed 1°C for extended periods. Do you have any idea why this would be the case, given that generally the differences are much smaller?

3. Comparing Figure 1 and Figure 3, with the exception of the southern portion of NEU the study areas seem almost to be orthogonal to the areas of maximum temperature anomaly, and one of the most striking AgPop regions where there is a high temperature anomaly, eastern Asia, is not included in the study. Thus the choice of study areas seems quite odd. It would surely be straightforward to include a relevant east Asian SREX region for completeness, which would mitigate the European/North American bias of this study.

4. In all three SREX regions of North America, the difference between the nudged run and either ERA-Interim or Berkeley for the maximum daily temperature anomalies

(Figure A4), especially for some of the largest values, can be much larger than the difference of the mean daily temperature anomalies (Figure 3). What is the reason for that? And how does it affect your estimates of extreme temperature? This feature suggests that using the climatological mean TX to bias-correct the TX values may not be adequate.

5. The left column of Figure 4 apparently includes a bias correction of the model output. This is only mentioned in the figure caption, not in the methods or anywhere else. Since the bias correction is almost certain to affect the results of the study, which are framed relative to a fixed temperature threshold of 40°C, a much more detailed assessment of its effect, and the potential error incurred thereby, is required. It appears that the bias correction was simply an adjustment of the mean, which assumes that the model TX distribution is perfect. Can you support this assumption with evidence? As noted in the previous comment, the assumption would appear to be contradicted by your own results. Why did you not use quantile mapping or some other more detailed method, which would treat the tails differently from the mean?

6. In lines 9-10, you should mention also the percentage value for the actual event, as a reference.

---

## Referee Comment (RC2) · Geert Jan van Oldenborgh (Referee) · 10 Jun 2020

Review of 'Storylines of the 2018 Northern Hemisphere heat wave at pre-industrial and higher global warming levels' by Kathrin Wehrli, Mathias Hauser, and Sonia I. Seneviratne.

This paper analyses what the heat waves of the NH summer of 2018 would look like in pre-industrial, current, 1.5 °C, 2 °C, 3 °C and 4 °C climates with the same circulation prescribed. This prescription makes it impossible to estimate probabilities, but the dependence of local heat wave temperatures and other properties such as drought and solar radiation can be shown, and a scaling with the global mean temperature

established. The results look solid and are certainly interesting, giving the important message that local temperature effects may be much stronger than the global mean temperature rise. It does not address the possible interconnectedness of the heat waves around the globe in that summer beyond citing Kornhuber (2019b).

There is only one major comment I have on the analysis, namely that it is only analyses climate model data and does not make any connection to observations beyond showing the patterns agree. Trends in heat waves are notoriously badly simulated by climate models and some comparisons of the modelled trends to the observed trends would make the paper and a discussion on possible discrepancies and how these would affect future trends much more useful for readers who want to apply the results to the real world rather than the model world.

As an example, I computed the observed trends corresponding to Fig.7 from CRU TS 4.04. The observed scaling factors are very different from the modelled ones, lower in North America and higher in Europe:

WNA: 0.9±0.2 K/K, CNA: 0.2±0.3 K/K (see eg https://doi.org/10.1038/s41467-020-16676-w), ENA: 0.6±0.2 K/K.

NEU: 1.2±0.3 K/K, CEU: 1.5±0.3 K/K, MED: 1.8±0.2 K/K.

With the addition of observed trends and a discussion on the differences with climate models (and the minor comments belo) the paper would be a useful contribution to the literature.

Minor comments

l.16 The Koreas were also very badly affected.

l.78 How does the end date of July 27 affect the results? Although this captures the largest area with heat, individual regions had heat waves after this date: North Korea experienced its worst heat the first days of August. The Benelux had a second heat wave in early August and the heat on the North American west coast was most severe

during the second week of August.

l.162 I would propose "almost simultaneous", there were weeks differences between these heat waves. Please also mention that there were severe heat waves after the cut-off date.

l.174 My Newfie friends prefer "Newfoundland".

l.201 Please mention that in contrast to the CMIP5 model simulations, observed precipitation has increased in CNA over the last century.

l.221 Why do you switch from a two-week period to a monthly period? The properties of short-duration heat waves are different from monthly anomalies. Please justify this choice.

---

## Author Comment (AC2) · 7 Jul 2020

*We thank Geert Jan for his evaluation of our manuscript and helpful comments, especially on observed trends. We have done additional analyses and update Figure 7 in the manuscript to follow the recommendations. Below we will answer the specific questions of the reviewer, addressing the major comment first and then the minor comments. For readability the questions are shown in* black *and answers are shown in* blue*.*

There is only one major comment I have on the analysis, namely that it is only analyses climate model data and does not make any connection to observations beyond showing

the patterns agree. Trends in heat waves are notoriously badly simulated by climate models and some comparisons of the modelled trends to the observed trends would make the paper and a discussion on possible discrepancies and how these would affect future trends much more useful for readers who want to apply the results to the real world rather than the model world.

We agree that observed trends should be discussed in the paper. Therefore, we made additional analyses using maximum daily temperatures for CRU TS4.03 and the Berkeley Earth Surface Temperature (BEST) project. As a reference for global mean temperature change (land+ocean) we used HADCRUT4 and GISTEMP. We estimated the slope using a linear regression for the years 1901-2017 from all data sets. The uncertainty of the fit for the slope is estimated using the covariance matrix. The results are shown in Figs. 1 and 2 given at the end of this comment, which will both be included in the manuscript (the latter in the appendix). The results indicate that, especially for CNA and ENA, the CMIP5 models overestimate the regional warming compared to observations, as documented in previous articles (e.g. Alter et al., 2017; Donat et al. 2017). Further articles also showed that the CMIP5 models tend to overestimate soil moisture-temperature coupling (Sippel et al. 2017, Vogel et al. 2018), which can lead to an overestimation of projected changes in temperature extremes (Vogel et al. 2018). These biases appear smaller in the newer CMIP6 models (Seneviratne and Hauser 2020). On the other hand, there can be large differences of the observed trend for some of the regions depending on the observational data sets used (e.g. MED and CEU; see Fig. 1 at the end of this comment). We discuss the systematic biases in CMIP5 models together with the differences and uncertainties of the observational data sets in the revised paper.

We are not sure which exact method the reviewer used to estimate observed scaling factors, i.e. which reference he used for global mean temperature, whether linear regression was used, which time periods were considered, and whether he included ocean grid points within the given SREX regions (which we do not for the regional

temperatures). Therefore, our results differ from the numbers given in the reviewer comment (see Fig. 2 at the end of this comment), although they agree on a general overestimation of regional warming per degree of global warming in the CMIP5 models for North America.

Both global temperature data sets (GISTEMP and HadCRUT4) merge near-surface temperatures over land with SSTs over the ocean, which leads to an inconsistency with how global mean temperature is commonly determined for models by taking near surface temperature over ocean and land (as is also done here; see also Cowtan et al., 2015). Therefore, 1 degree of global mean temperature increase in the observations does not correspond to 1 degree from the models (see also IPCC, 2018). We discuss this issue in the paper.

References:
Alter, R. E., Douglas, H. C., Winter, J. M., Eltahir, E. A. B. (2018). Twentieth century regional climate change during the summer in the central United States attributed to agricultural intensification. Geophysical Research Letters, 45, 1586-1594. https://doi.org/10.1002/2017GL075604.

Cowtan, K., Hausfather, Z., Hawkins, E., Jacobs, P., Mann, M. E., Miller, S. K., Steinman, B. A., Stolpe, M. B., and Way, R. G. (2015), Robust comparison of climate models with observations using blended land air and ocean sea surface temperatures, Geophys. Res. Lett., 42, 6526-6534, https://doi.org/10.1002/2015GL064888.

Donat, M.G., A.J. Pitman, and S.I. Seneviratne (2017). Regional warming of hot extremes accelerated by surface energy fluxes, Geophys. Res. Lett., 44, https://doi.org/10.1002/2017GL073733.

IPCC, 2018: Summary for Policymakers. In: Global Warming of 1.5°C. An IPCC Special Report on the impacts of global warming of 1.5°C above pre-industrial levels and related global greenhouse gas emission pathways, in the context of strengthening the global response to the threat of climate change, sustainable

development, and efforts to eradicate poverty [Masson-Delmotte, V., P. Zhai, H.-O. P.rtner, D. Roberts, J. Skea, P. R. Shukla, A. Pirani, W. Moufouma-Okia, C. P.an, R. Pidcock, S. Connors, J. B. R. Matthews, Y. Chen, X. Zhou, M.I. Gomis, E. Lonnoy, T. Maycock, M. Tignor, and T. Waterfield (eds.)]. In Press. Available from https://www.ipcc.ch/site/assets/uploads/sites/2/2019/05/SR15_SPM_version_report_LR.pdf.

Seneviratne, S. I., and M. Hauser (2020). Regional climate sensitivity of climate extremes in CMIP6 vs CMIP5 multi-model ensembles. Earth's Future, https://doi.org/10.1029/2019EF001474.

Sippel, S., J. Zscheischler, M. D. Mahecha, R. Orth, M. Reichstein, M. Vogel and S. I. Seneviratne (2017). Refining multi-model projections of temperature extremes by evaluation against land-atmosphere coupling diagnostics. Earth Syst. Dynam., 8, 387-403, https://doi.org/10.5194/esd-8-387-2017.

Vogel, M. M., J. Zscheischler, and S. I. Seneviratne (2018). Varying soil moisture-atmosphere feedbacks explain divergent temperature extremes and precipitation projections in central Europe. Earth Syst. Dynam., 9, 1107-1125, https://doi.org/10.5194/esd-9-1107-2018.

**Minor comments**
l.16 The Koreas were also very badly affected.

We included the Korean Peninsula in the list of affected regions.

l.78 How does the end date of July 27 affect the results? Although this captures the largest area with heat, individual regions had heat waves after this date: North Korea experienced its worst heat the first days of August. The Benelux had a second heatwave in early August and the heat on the North American west coast was most severe

during the second week of August.

We agree that with the presented study we cannot make statements about heat waves after 27 July 2018, which were more intense in some locations. The choice was also made due to the availability of input files for the atmospheric nudging. Therefore, we cannot provide numbers on how the results are affected by the chosen time period. We changed the text in the results and discussion as well as the conclusions sections of the manuscript to mention the regions affected by heat waves after July 27 and to discuss this shortcoming of our study.

l.162 I would propose "almost simultaneous", there were weeks differences between these heat waves. Please also mention that there were severe heat waves after the cut-off date.

We agree and use the term "almost simultaneous" as suggested. Also we mention that some regions experienced severe heat waves after the cut-off date (see also previous answer).

l.174 My Newfie friends prefer "Newfoundland".

We thank the reviewer for spotting and correcting this.

l.201 Please mention that in contrast to the CMIP5 model simulations, observed precipitation has increased in CNA over the last century.

We added that the simulated trends are of different sign than the observed summer precipitation trends for CNA.

l.221 Why do you switch from a two-week period to a monthly period? The properties of short-duration heat waves are different from monthly anomalies. Please justify this

choice.

For the CMIP5 models the monthly period was chosen for two reasons: First, as the CMIP5 models do not have a prescribed ocean or nudged atmosphere, there is more variability which is better represented using a longer sample and it is also not necessary to have the exact days as every day of July matches conditions of the study time period. Secondly, it is more practicable to process the monthly instead of the daily data. For the nudged experiments in our study we agree that the choice of the time period and its length could have been expected to affect the results, although we find that the effects are actually very small. To assess this point, we repeated the analysis plotting only July 13-27 for CESM nudged. The results show that changing the monthly to a two-week period for the nudged experiments does not substantially impact the results (see Fig. 3 at the end of this comment). The change is largest for ENA where a reduction of the slope can be found. However, results are still qualitatively the same.

[Figure]

**Fig. 1.** As Fig. 7 in the paper but added observed trends. The solid green lines correspond to the approximate observed warming while dashed green lines indicate the extrapolation beyond the observed warming.

| TXreg | CRUTS | | BEST | |
|---|---|---|---|---|
| **Tglob** | **GISTEMP** | **HadCRUT** | **GISTEMP** | **HadCRUT** |
| WNA | 0.85 ± 0.21 | 1.03 ± 0.24 | 0.85 ± 0.22 | 1.04 ± 0.25 |
| CNA | 0.16 ± 0.30 | 0.24 ± 0.34 | 0.55 ± 0.31 | 0.68 ± 0.35 |
| ENA | 0.56 ± 0.22 | 0.65 ± 0.25 | 0.58 ± 0.21 | 0.68 ± 0.23 |
| CEU | 1.52 ± 0.30 | 1.84 ± 0.33 | 1.30 ± 0.30 | 1.60 ± 0.34 |
| NEU | 1.13 ± 0.31 | 1.32 ± 0.35 | 1.11 ± 0.29 | 1.29 ± 0.33 |
| MED | 1.67 ± 0.15 | 1.93 ± 0.16 | 1.21 ± 0.16 | 1.40 ± 0.18 |
| EAS | 0.61 ± 0.16 | 0.80 ± 0.18 | 0.55 ± 0.19 | 0.75 ± 0.21 |
| AgPop | 0.72 ± 0.12 | 0.87 ± 0.14 | 0.80 ± 0.13 | 0.96 ± 0.14 |

**Fig. 2.** The slopes for all combinations of the observational data sets and their uncertainties (one standard deviation).

[Figure]

**Fig. 3.** Same as Fig. 7 in the paper but showing July 13-27 for the nudged simulations (dark blue; stippled) and July 1-27 (dark blue; solid).

---

## Author Response (AR1)

Dear editor

We have considered all the major and minor comments by the reviewers and revised the manuscript accordingly. Following relevant changes were done to the manuscript:

- More accurate bias correction of the absolute values for daily maximum temperature (TX) using quantile-mapping (shown in updated Figure 4a)
- The bias corrected TX and hence, values for percentage of AgPop region affected by extremely hot temperatures, have changed due to the new bias correction method
- Added verification of bias corrected TX for 2018 using observational data sets and reanalysis (e.g. Fig. A3)
- Inclusion of the Eastern Asia SREX region in the figures and throughout the text
- Added discussion of model biases and uncertainties in observations of regional TX trends
- New Table A1 in the appendix showing observed regional TS trends and their uncertainties using two global mean temperature data sets (GISTEMPv4, HadCRUT4) and two regional land temperature data sets (CRU TSv4.03, Berkeley Earth)
- Included observed TX trends in Fig. 7
- Updated Fig. 7 also to show the study time period (13-27 July) for the nudged CESM simulations
- Clarification of the choice for the end date for the study period and the nudging input files as well as expanded discussion of the implications for regions that were affected by heat waves after this end date

Please find below a point-by-point reply to the comments made by the reviewers and a marked up manuscript version showing the changes made.

Sincerely
Kathrin Wehrli (on behalf of all authors)

**Response to Anonymous Referee #1**

The concept of asking how a given meteorological event might have been exacerbated by global warming has been growing in popularity as an alternative to the more common approach of probabilistic event attribution. As first proposed by Trenberth et al.(2015 doi: 10.1038/NCLIMATE2657) and Shepherd (2016 doi: 10.1007/s40641-016-0033-y), this 'storyline' approach takes the atmospheric flow configuration leading to the event as given, and quantifies the impact of global warming conditional on that flow configuration. The arguments for why this may be useful are given in those two papers,but the general concept is that generality is sacrificed in order to obtain a more detailed and hopefully more informative statement of impact (since tied to a particular event).The storyline approach was first applied to synoptic-timescale weather phenomena, e.g. tropical cyclones, where the conditionality was applied either through the initial conditions in a forecast, or through the boundary conditions in a regional model. Here the storyline approach is applied to a multi-week heat wave event, nudging the circulation in a global model to reanalysis, following the methodology previously used by the authors in their 2019 JGR paper to understand the role of soil-moisture feedbacks in heat waves. It is important to document applications of the storyline approach in different contexts so that we can learn to understand its strengths and weaknesses. From that perspective this study is welcome, and for the most part the results are carefully explained and clearly presented. I find Figure 7 to be the most interesting of all. I am happy to recommend publication, provided the following points are addressed:

A1: We thank the reviewer for the positive and thoughtful evaluation of the manuscript. We appreciate the comments on the selection of the study regions and the bias correction of absolute maximum daily temperatures. We have followed the recommendations and computed new figures that will be included and discussed in the revised manuscript. We now use quantile-mapping to bias-correct our model simulations. The area affected by maximum daily temperature > 40°C has changed due to the new bias correction method. The new results agree better with observations and are qualitatively still in-line with earlier results. Below we will answer the specific questions of the reviewer.

1. Figure 3 shows only temperature anomalies. It would be good to also show absolute temperatures (e.g. in the maps), so that the reader can see the extent of the temperature bias of the model.

A2: We agree with the reviewer that biases of absolute temperature of the model and their correction is important. It is well-known that the majority of CMIP5 models, including CESM, overestimate summer temperatures in Northern Hemisphere midlatitudes (e.g. Mueller and Seneviratne, 2014, GRL; Wehrli et al., 2018, GRL; also shown for TXx in CESM in the latter). In the revised manuscript we add plots of absolute TX (mean over Jul. 13-27 2018) to Appendix Figure A3, which show the bias-corrected (using quantile mapping) CESM historical simulation against different references. Since it is known that model biases are large, we do not think it is necessary to show the magnitude of the bias. It is, however, crucial to discuss the bias correction and check the bias-corrected model against reference data sets. For reference we show a comparison of TXx > 40°C for the original model output, mean bias correction and quantile mapping (using both Berkeley-Earth and ERA-Interim as reference) against ERA-Interim and Berkeley Earth (see attachment/below).

References:
Mueller, B., and Seneviratne, S. I. (2014), Systematic land climate and evapotranspiration biases in CMIP5 simulations, *Geophys. Res. Lett.*, 41, 128-134, doi:10.1002/2013GL058055.

Wehrli, K., Guillod, B. P., Hauser, M., Leclair, M., & Seneviratne, S. I. (2018). Assessing the dynamic versus thermodynamic origin of climate model biases. *Geophysical Research Letters*, 45, 8471-8479. https://doi.org/10.1029/2018GL079220

2. In some periods and regions, the differences between the nudged run and ERA-Interim anomalies in the time series in Figure 3 can exceed 1°C for extended periods. Do you have any idea why this would be the case, given that generally the differences are much smaller?

A3: We can only speculate about the reason for the large differences between the nudged run and ERA-Interim temperature anomalies in Figure 3. The most striking case is NEU and we verified that the difference is largest during two shorter periods in mid- and end of June and the period shown in Figure 3 in the beginning of July. For these periods warm anomalies are overestimated, whereas during the rest of the year the differences are generally smaller. One possibility is that during the NH2018 heatwave soil moisture got depleted even in regions that are usually rich in moisture (so-called wet regime), which causes the land surface to react very sensitive to a further decrease in soil moisture and to incoming radiation (i.e. change from wet to transitional regime). If the model dries faster or transitions to a radiation-sensitive state earlier than ERA-Interim this might result in a more sensitive and more pronounced response in temperature. With decreasing moisture availability more incoming radiation will contribute to sensible heat flux and hence to increased temperature.

3. Comparing Figure 1 and Figure 3, with the exception of the southern portion of NEU the study areas seem almost to be orthogonal to the areas of maximum temperature anomaly, and one of the most striking AgPop regions where there is a high temperature anomaly, eastern Asia, is not included in the study. Thus the choice of study areas seems quite odd. It would surely be straightforward to include a relevant east Asian SREX region for completeness, which would mitigate the European/North American bias of this study.

A4: We agree with the reviewer that our choice of regions was biased towards Europe and North America. In the revised manuscript we include the Eastern Asian SREX region (EAS) in the figures and analysis. The region of Neufundland/Québec would also be interesting to examine. However, there is no SREX region that would be suitable. The Canada/ Greenland/ Iceland SREX region (CGI) encompasses large areas with temperature anomalies of the opposite sign (e.g. Greenland). We decided to not define a new region specifically for this case.
Apart from the just-mentioned region in north-eastern America, we believe that the interesting regions for the 2018 heat wave are addressed in this study. The Mediterranean was not strongly affected by heat waves during summer 2018, which was discussed in other studies (e.g. Toreti et al., 2019). Therefore, we thought it is interesting to include this region in the analysis. Except for the scaling plots (Figure 7) and time series in Figure 3, results always show the entire Northern Hemisphere north of 25°N and the discussion is not limited to the SREX regions.

Reference:
Toreti, A., Belward, A.,Perez-Dominguez, I., Naumann, G., Luterbacher, J., Cronie, O., et al. (2019). The exceptional 2018 European water seesaw calls for action on adaptation. *Earth's Future*, 7, 652–663. https://doi.org/10.1029/2019EF001170

4. In all three SREX regions of North America, the difference between the nudged run and either ERA-Interim or Berkeley for the maximum daily temperature anomalies (Figure A4), especially for some of the largest values, can be much larger than the difference of the mean daily temperature anomalies (Figure 3). What is the reason for that? And how does it affect

your estimates of extreme temperature? This feature suggests that using the climatological mean TX to bias-correct the TX values may not be adequate.

A5: One reason why differences of maximum temperature are larger than differences of mean daily temperature is that in the latter biases during the night can be balanced out by biases during the day and vice versa (overestimated/underestimated diurnal cycle can still show as correct daily mean). Connected to that, biases of mean daily temperature anomalies can affect biases of extreme temperatures (and vice versa) but they don't have to in a direct, linear way. We agree that the bias-correction of TX should be treated in more details and several methods should be evaluated for their adequacy. In the revised manuscript we apply a quantile-mapping that was tested for two reference data sets (see next answer A6).

5. The left column of Figure 4 apparently includes a bias correction of the model output. This is only mentioned in the figure caption, not in the methods or anywhere else. Since the bias correction is almost certain to affect the results of the study, which are framed relative to a fixed temperature threshold of 40°C, a much more detailed assessment of its effect, and the potential error incurred thereby, is required. It appears that the bias correction was simply an adjustment of the mean, which assumes that the model TX distribution is perfect. Can you support this assumption with evidence? As noted in the previous comment, the assumption would appear to be contradicted by your own results. Why did you not use quantile mapping or some other more detailed method, which would treat the tails differently from the mean?

A6: We agree with the reviewer that the bias correction was not addressed sufficiently in the manuscript. The presented method was a day-of-year dependent correction of the mean (TX) bias. We tested the presented method against a quantile-mapping bias correction using a 91-day moving window (hence, making it also dependent on the day-of-year). As could be expected, the quantile-mapped results reveal that the bias correction method strongly influences the results and the mean bias correction is less appropriate in our case. We discuss this in the revised manuscript. Figure 4 and the numbers in the manuscript are replaced by the results from the quantile-mapping. We verified that the quantile mapping leads to better results: TX RMSE for the study period (Jul. 13-27 2018) and all land areas north of 25°N is reduced from 7.48°C in the original model to 1.95°C using mean bias correction and to 1.45°C using quantile mapping (Berkeley-Earth as reference; qualitatively the same is true for the RMSE of the AgPop region). The area affected by temperatures > 40°C discussed in the conclusion better match the reference data sets (Berkeley, ERA-Interim, MERRA-2) when using the quantile mapping. Below we included a figure (Figure 1) to show the effect of bias correction on TXx. The values for the AgPop region > 40°C are 9.1% and 8.5% for ERA-Interim and Berkeley-Earth, respectively. The bias corrected model simulates 8.8% and 9.3% area affected (depending on the reference used for the calibration of the quantile mapping), which is more accurate than the 20% we obtain for the mean bias correction.

[Figure]

Figure 1: TXx for 13-27 July 2018 and fraction of the AgPop region experiencing maximum daily temperatures larger than 40°C for the original and bias-corrected model output as well as two reference data sets.

6. In lines 9-10, you should mention also the percentage value for the actual event, as a reference.

A7: We agree and adjusted the last line of the abstract to include the percentage value for the actual event (now changed to the new value from the quantile-mapped simulations).

**Response to Geert Jan van Oldenborgh**

This paper analyses what the heat waves of the NH summer of 2018 would look like in pre-industrial, current, 1.5°C, 2°C, 3°C and 4°C climates with the same circulation prescribed. This prescription makes it impossible to estimate probabilities, but the dependence of local heat wave temperatures and other properties such as drought and solar radiation can be shown, and a scaling with the global mean temperature established. The results look solid and are certainly interesting, giving the important message that local temperature effects may be much stronger than the global mean temperature rise. It does not address the possible interconnectedness of the heatwaves around the globe in that summer beyond citing Kornhuber (2019b). There is only one major comment I have on the analysis, namely that it is only analyses climate model data and does not make any connection to observations beyond showing the patterns agree. Trends in heat waves are notoriously badly simulated by climate models and some comparisons of the modelled trends to the observed trends would make the paper and a discussion on possible discrepancies and how these would affect future trends much more useful for readers who want to apply the results to the real world rather than the model world. As an example, I computed the observed trends corresponding to Fig.7 from CRU TS4.04. The observed scaling factors are very different from the modelled ones, lower in North America and higher in Europe:
WNA: 0.9±0.2 K/K, CNA: 0.2±0.3 K/K (see eg https://doi.org/10.1038/s41467-020-16676-w), ENA: 0.6±0.2 K/K.
NEU: 1.2±0.3 K/K, CEU: 1.5±0.3 K/K, MED: 1.8±0.2 K/K.
With the addition of observed trends and a discussion on the differences with climate models (and the minor comments below) the paper would be a useful contribution to the literature.

B1: We thank Geert Jan van Oldenborgh for his comment on observed trends and agree that it should be discussed in the paper. Therefore, we made additional analyses using maximum daily temperatures for CRU TS4.03 and the Berkeley Earth Surface Temperature (BEST) project. As a reference for global mean temperature change (land+ocean) we used HADCRUT4 and GISTEMP. We estimated the slope using a linear regression for the years 1901-2017 for all combinations of the 4 data sets. The uncertainty of the fit for the slope is estimated using the covariance matrix.
The results are shown in the figure and table given below, which will both be included in the manuscript (the table in the appendix). The results indicate that, especially for CNA and ENA, the CMIP5 models overestimate the regional warming compared to observations, as documented in previous articles (e.g. Alter et al., 2017; Donat et al. 2017). Further articles also showed that the CMIP5 models tend to overestimate soil moisture-temperature coupling (Sippel et al. 2017, Vogel et al. 2018), which can lead to an overestimation of projected changes in temperature extremes (Vogel et al. 2018). These biases appear smaller in the newer CMIP6 models (Seneviratne and Hauser 2020). On the other hand, there can be large differences of the observed trend for some of the regions depending on the observational data sets used (e.g. MED and CEU; see Figure 2 below). We discuss the systematic biases in CMIP5 models together with the differences and uncertainties of the observational data sets in the paper.

We are not sure which exact method the reviewer used to estimate observed scaling factors, i.e. which reference he used for global mean temperature, whether linear regression was used, which time periods were considered, and whether he included ocean grid points within the given SREX regions (which we do not for the regional temperatures). Therefore, our results differ from the numbers given in the reviewer comment (see Table 1 below), although they agree on a general overestimation of regional warming per degree of global warming in the CMIP5 models for North America.

Both global temperature data sets (GISTEMP and HadCRUT4) merge near-surface temperatures over land with SSTs over the ocean, which leads to an inconsistency with how global mean temperature is commonly determined for models by taking near surface temperature over ocean and land (as is also done here; see also Cowtan et al., 2015). Therefore, 1 degree of global mean temperature increase in the observations does not correspond to 1 degree from the models (see also IPCC, 2018). We discuss this issue in the paper.

[Figure]

Figure 2: As Fig. 7 in the paper but added observed trends. The solid green lines correspond to the approximate observed warming while dashed green lines indicate the extrapolation beyond the observed warming.

Table 1: The slopes for all combinations of the observational data sets and their uncertainties (one standard deviation).

| TXreg | CRUTS | | BEST | |
|---|---|---|---|---|
| Tglob | GISTEMP | HadCRUT | GISTEMP | HadCRUT |
| WNA | 0.85 ± 0.21 | 1.03 ± 0.24 | 0.85 ± 0.22 | 1.04 ± 0.25 |
| CNA | 0.16 ± 0.30 | 0.24 ± 0.34 | 0.55 ± 0.31 | 0.68 ± 0.35 |
| ENA | 0.56 ± 0.22 | 0.65 ± 0.25 | 0.58 ± 0.21 | 0.68 ± 0.23 |
| CEU | 1.52 ± 0.30 | 1.84 ± 0.33 | 1.30 ± 0.30 | 1.60 ± 0.34 |
| NEU | 1.13 ± 0.31 | 1.32 ± 0.35 | 1.11 ± 0.29 | 1.29 ± 0.33 |
| MED | 1.67 ± 0.15 | 1.93 ± 0.16 | 1.21 ± 0.16 | 1.40 ± 0.18 |
| EAS | 0.61 ± 0.16 | 0.80 ± 0.18 | 0.55 ± 0.19 | 0.75 ± 0.21 |
| AgPop | 0.72 ± 0.12 | 0.87 ± 0.14 | 0.80 ± 0.13 | 0.96 ± 0.14 |

**Minor comments**

l.16 The Koreas were also very badly affected.

B2: We included the Korean Peninsula in the list of affected regions.

l.78 How does the end date of July 27 affect the results? Although this captures the largest area with heat, individual regions had heat waves after this date: North Korea experienced its worst heat the first days of August. The Benelux had a second heatwave in early August and the heat on the North American west coast was most severe during the second week of August.

B3: We agree that with the presented study we cannot make statements about heat waves after 27 July 2018, which were more intense in some locations. The choice was also made due to the availability of input files for the atmospheric nudging. Therefore, we cannot provide numbers on how the results are affected by the chosen time period. We changed the text in the results and discussion as well as the conclusions of the manuscript to mention the regions affected by heat waves after July 27 and to discuss this shortcoming of our study.

l.162 I would propose "almost simultaneous", there were weeks differences between these heat waves. Please also mention that there were severe heat waves after the cut-off date.

B4: We agree and use the term "almost simultaneous" as suggested. Also we mention that some regions experienced severe heat waves after the cut-off date (see also answer B3).

l.174 My Newfie friends prefer "Newfoundland".

B5: We thank the reviewer for spotting and correcting this.

l.201 Please mention that in contrast to the CMIP5 model simulations, observed precipitation has increased in CNA over the last century.

B6: We added that the simulated trends are of different sign than the observed summer precipitation trends for CNA.

l.221 Why do you switch from a two-week period to a monthly period? The properties of short-duration heat waves are different from monthly anomalies. Please justify this choice.

B7: For the CMIP5 models the monthly period was chosen for two reasons: First, as the CMIP5 models do not have a prescribed ocean or nudged atmosphere, there is more variability which is better represented using a longer sample and it is also not necessary to have the exact days as every day of July matches conditions of the study time period. Secondly, it is more practicable to process the monthly instead of the daily data. For the nudged experiments in our study we agree that the choice of the time period and its length could have been expected to affect the results, although we find that the effects are actually very small. To assess this point, we repeated the analysis plotting only July 13-27 for CESM nudged. The results show that changing the monthly to a two-week period for the nudged experiments does not substantially impact the results (see Figure 3 below). The change is largest for ENA where a reduction of the slope can be found. However, results are still qualitatively the same.

[Figure]

Figure 3: Same as Figure 7 in the paper but showing July 13-27 for the nudged simulations (dark blue; stippled) and July 1-27 (dark blue; solid).

[revised manuscript text omitted]